# A Differentially Private Clustering Algorithm for Well-Clustered Graphs

**Weiqiang He**[1]    **Hendrik Fichtenberger**[2]    **Pan Peng**[1]*

[1]School of Computer Science and Technology, University of Science and Technology of China
[2]Google Research

hwqhwq@mail.ustc.edu.cn   fichtenberger@google.com   ppeng@ustc.edu.cn

## Abstract

We study differentially private (DP) algorithms for recovering clusters in well-clustered graphs, which are graphs whose vertex set can be partitioned into a small number of sets, each inducing a subgraph of high inner conductance and small outer conductance. Such graphs have widespread application as a benchmark in the theoretical analysis of spectral clustering. We provide an efficient $(\epsilon,\delta)$-DP algorithm tailored specifically for such graphs. Our algorithm draws inspiration from the recent work of Chen et al. [NeurIPS'23], who developed DP algorithms for recovery of stochastic block models in cases where the graph comprises exactly two nearly-balanced clusters. Our algorithm works for well-clustered graphs with $k$ nearly-balanced clusters, and the misclassification ratio almost matches the one of the best-known non-private algorithms. We conduct experimental evaluations on datasets with known ground truth clusters to substantiate the prowess of our algorithm. We also show that any (pure) $\epsilon$-DP algorithm would result in substantial error.

## 1 Introduction

Graph Clustering is a fundamental task in unsupervised machine learning and combinatorial optimization, relevant to various domains of computer science and their diverse practical applications. The goal of Graph Clustering is to partition the vertex set of a graph into distinct groups (or clusters) so that similar vertices are grouped in the same cluster while dissimilar vertices are assigned to different clusters.

There exist numerous notions of similarity and measures of evaluating the quality of graph clusterings, with *conductance* being one of the most extensively studied (see e.g. (Kannan et al., 2004; Von Luxburg, 2007; Gharan & Trevisan, 2012)). Formally, let $G = (V, E)$ be an undirected graph. For any vertex $u \in V$, its degree is denoted by $\mathrm{d}_G(u)$, and for any set $S \subseteq V$, its *volume* is $\mathrm{vol}_G(S) = \sum_{u \in S} \mathrm{d}_G(u)$. For any two subsets $S, T \subset V$, we define $E(S, T)$ to be the set of edges between $S$ and $T$. For any nonempty subset $C \subset V$, the *outer conductance* and *inner conductance* are defined by

$$\Phi_{\text{out}}(G, C) := \frac{|E(C, V \setminus C)|}{\mathrm{vol}_G(C)}, \quad \Phi_{\text{in}}(G, C) := \min_{S \subseteq C, \mathrm{vol}_G(S) \leq \frac{\mathrm{vol}_G(C)}{2}} \Phi_{\text{out}}(C, S)$$

Intuitively, if a vertex set $C$ has low outer conductance, then it has relatively few connections to the outside, and if it has high inner conductance, then it is well connected inside. Based on this intuition, Gharan & Trevisan (2014) introduced the following notion of *well-clustered graphs*. A *k-partition* of a graph $G = (V, E)$ is a family of $k$ disjoint vertex subsets $C_1, \ldots, C_k$ of $V$ such that the union $\cup_{i=1}^{k} C_i = V$. If there is some constant $c \in (0, 1]$ such that for every $i \in [k]$, $\mathrm{vol}_G(C_i) \geq \frac{c\mathrm{vol}_G(G)}{k} = \frac{2cm}{k}$ is satisfied, we call the $k$-partition $\{C_i\}_{i \in [k]}$ *c-balanced*.

**Definition 1.1** (Well-clustered graph). *Given parameters $k \geq 1$, $\phi_{in}, \phi_{out} \in [0, 1]$, a graph $G = (V, E)$ is called $(k, \phi_{\text{in}}, \phi_{\text{out}})$-clusterable if there exists a k-partition $\{C_i\}_{i \in [k]}$ of $V$ such that for*

---

*Corresponding Author

all $i \in [k]$, $\Phi_{in}(G, C_i) \geq \phi_{in}$ and $\Phi_{out}(G, C_i) \leq \phi_{out}$. *Furthermore, if $\{C_i\}_{i \in [k]}$ is c-balanced, $G = (V, E)$ is called c-balanced $(k, \phi_{in}, \phi_{out})$-clusterable.*

If a $k$-partition $\{C_i\}_{i \in [k]}$ satisfies the above two conditions on the inner and outer conductances, then we call the partition a *ground truth partition* of $G$. Gharan & Trevisan (2014) give a simple polynomial-time spectral algorithm to approximately find such a partitioning. Since then, a plethora of works has focused on extracting the cluster structure of such graphs with spectral methods (see Section 1.1). For example, Czumaj et al. (2015); Peng et al. (2015); Chiplunkar et al. (2018); Peng (2020); Gluch et al. (2021) used well-clustered graphs as a theoretical arena to gain a better understanding of why *spectral clustering* is successful. They showed that variants of the widely used spectral clustering give a good approximation of the optimal clustering in a well-clustered graph.

In this paper, we study *differentially private (DP)* (see Definition 2.1 for the formal definition) algorithms for recovering clusters in well-clustered graphs. DP algorithms aim to enable statistical analyses of sensitive information on individuals while providing strong guarantees that no information of any individual is leaked (Dwork et al., 2006). Given the success of spectral methods for clustering graphs in non-private settings, surprisingly little is known about differentially private spectral clustering. Finding ways to leverage these methods in a privacy-preserving way gradually bridges differential privacy and an area of research with many deep structural results and a strong toolkit for graphs. In this line of research, we obtain the following result.

**Theorem 1.** *Let $G = (V, E)$ be a c-balanced $(k, \phi_{in}, \phi_{out})$-clusterable graph with its ground truth partition $\{C_i\}_{i \in [k]}$, where $\frac{\phi_{out}}{\phi_{in}^2} = O(k^{-4})$. Then, there exists an algorithm that, for any $c, k, \phi_{in}, \phi_{out}$ and graph $G$ with $n$ vertices and $m \geq n \cdot \frac{\phi_{in}^4}{\phi_{out}^2} \cdot \frac{\log(2/\delta)}{\epsilon^2}$ edges that satisfies the preceding properties, outputs a $k$-partition $\{\widehat{C}_i\}_{i=1}^k$ such that*

$$\mathrm{vol}_G(\widehat{C}_i \triangle C_{\sigma(i)}) = O\left(\frac{k^4}{c^2} \cdot \frac{\phi_{out}}{\phi_{in}^2}\right) \cdot \mathrm{vol}_G(C_{\sigma(i)}), \quad \text{for any } i \leq k$$

*with probability $1 - \exp(-\Omega(n))$, where $\sigma$ is a permutation over $[k] := \{1, \ldots, k\}$. Moreover, the algorithm is $(\epsilon, \delta)$-DP for any input graph with respect to edge privacy and runs in polynomial time.*

For a more general trade-off between the parameters $\epsilon, \delta$ and the misclassification ratio of our algorithm, we refer to Lemma 4.3. We stress that the privacy guarantee of our algorithms holds for *any* input graph, and in particular, it does not depend on a condition that the graph is clusterable. In the non-private setting, the best-known efficient algorithms achieve $O(k^3 \cdot \phi_{out}/\phi_{in}^2)$ misclassification ratio (for general well-clustered graphs) under the assumption that[1] $\phi_{out}/\phi_{in}^2 = O(k^{-3})$ (Peng et al., 2015). Thus, for balanced well-clustered graphs, our private algorithm almost matches the best-known non-private one in terms of approximate accuracy or utility.

Our private mechanism is inspired by the recent work of Chen et al. (2023). We design a simple *Semi-Definite Program (SDP)* and run spectral clustering on a noisy solution. To analyze our algorithm, we extend the notion of *strong convexity* and prove the stability of the SDP. This allows us to show that the solution of the SDP has small *sensitivity*. In differential privacy, sensitivity is a measure of how much a function's value changes for small, but arbitrary and possibly adversarial changes in the data. For the non-private SDP solution, we show that applying classical privacy mechanisms and spectral clustering yields a differentially private clustering algorithm. Based on the analysis by Peng et al. (2015), we prove that our differentially private clustering algorithm achieves an approximate accuracy that nearly matches the non-private version. Furthermore, we remark that Chen et al. (2023) give a DP algorithm for recovery of *stochastic block model (SBM)* in cases where the graph comprises exactly two nearly-balanced clusters. Our algorithm achieves a similar approximation accuracy to their *weak recovery* and supports $k$ clusters.

To complement our results, we conduct an experimental evaluation on datasets with known ground truth clusters to substantiate the prowess of our algorithm. Furthermore, we show that any (pure) $\epsilon$-DP algorithm entails substantial error in its output (see Appendix D).

**Theorem 2** (informal). *Any algorithm for the cluster recovery of well-clustered graphs with failure probability $\eta$ and misclassification rate $\zeta$ cannot satisfy $\epsilon$-DP for $\epsilon < \frac{2\ln(\frac{1}{9\epsilon\zeta})}{d}$ on d-regular graphs.*

---

[1]Strictly speaking, Peng et al. (2015) stated their result in terms of a so-called quantity $\Upsilon$, which can be lower bounded by $\phi_{in}^2/\phi_{out}$ by higher Cheeger inequality (Lee et al., 2014).

This lower bound implies that, e.g., there is no $\epsilon$-DP algorithm for $\epsilon \in \Omega(1)$ that misclassifies only a constant fraction of the input. On the other hand, Theorem 1 confirms the existence of such an algorithm for $(\epsilon, \delta)$-DP.

## 1.1 RELATED WORK

Our results combine spectral clustering and differential privacy. After Dwork et al. (2006) systematically introduced the notion of differential privacy, various clustering objectives have been studied in this regime. Differentially private metric clustering (e.g., $k$-clustering) was studied in Nissim et al. (2007); Wang et al. (2015); Huang & Liu (2018); Shechner et al. (2020); Ghazi et al. (2020). Correlation Clustering with differential privacy has been the subject of more recent works, e.g., Bun et al. (2021); Liu (2022); Cohen-Addad et al. (2022). In machine learning, the SBM and mixture models are used to model and approximate properties of real graphs. Differentially private recovery of these graphs or their properties has been studied in Hehir et al. (2022); Mohamed et al. (2022); Chen et al. (2023); Seif et al. (2023).

*Spectral* methods for graphs have a long track of research that started before differential privacy was widely studied. One of the first theoretical analyses of these methods appeared in Spielman & Teng (1996), followed by several others, e.g., Von Luxburg (2007); Gharan & Trevisan (2014); Peng et al. (2015); Kolev & Mehlhorn (2016); Dey et al. (2019); Mizutani (2021). Indeed, there has been a long line of research aiming at designing efficient algorithms for extracting clusters with small outer conductance (and with high inner conductance) in a graph (Spielman & Teng, 2004; Andersen et al., 2006; Gharan & Trevisan, 2012; Zhu et al., 2013). Privacy-preserving spectral methods have been studied rather recently (Wang et al. (2013); Arora & Upadhyay (2019); Cui et al. (2021)).

For *well-clustered* graphs, *spectral clustering* is the most commonly used and effective algorithm, which operates in the following two steps: (1) construct a *spectral embedding*, mapping vertices into a $k$-dimensional real space. (2) use $k$-means or other algorithms to cluster the points in Euclidean space. Spectral clustering has been applied in many fields and has achieved significant results (Alpert & Yao, 1995; Shi & Malik, 2000; Ng et al., 2001; Malik et al., 2001; Belkin & Niyogi, 2001; Liu & Zhang, 2004; Zelnik-Manor & Perona, 2004; White & Smyth, 2005; Von Luxburg, 2007; Wang & Dong, 2012; Taşdemir, 2012; Cucuringu et al., 2016).

## 2 PRELIMINARIES

We use bold symbols to represent vectors and matrices. For vectors $\boldsymbol{u}, \boldsymbol{v}$, we define $\langle \boldsymbol{u}, \boldsymbol{v} \rangle := \boldsymbol{u}^\top \boldsymbol{v} = \sum_i \boldsymbol{u}_i \boldsymbol{v}_i$. We denote by $\|\boldsymbol{u}\|_1 := \sum_i |\boldsymbol{u}_i|$, $\|\boldsymbol{u}\|_2 := \sqrt{\sum_i \boldsymbol{u}_i^2}$ its $\ell_1$ norm and $\ell_2$ norm, respectively. For matrices $\boldsymbol{A}$ and $\boldsymbol{B}$, define $\langle \boldsymbol{A}, \boldsymbol{B} \rangle := \sum_{i,j} \boldsymbol{A}_{ij} \boldsymbol{B}_{ij}$. Denote by $\|\boldsymbol{A}\|_2$ the spectral norm of $\boldsymbol{A}$. Denote by $\|\boldsymbol{A}\|_F$ the Frobenius norm of $\boldsymbol{A}$. A matrix $\boldsymbol{A}$ is said to be *positive semidefinite* if there is a matrix $\boldsymbol{V}$ such that $\boldsymbol{A} = \boldsymbol{V}^\top \boldsymbol{V}$, denoted as $\boldsymbol{A} \succeq 0$. Note that $\boldsymbol{A}_{ij} = \boldsymbol{v_i} \cdot \boldsymbol{v_j}$, where $\boldsymbol{v_i}$ is the $i$-th column of $\boldsymbol{V}$ and $\boldsymbol{A}$ is known as the *Gram matrix* of these vectors $\boldsymbol{v_i}, i \in [n]$. For $n \geq 1$, let $[n] = \{1, \ldots, n\}$. Let $\textbf{Diag}(a_1, a_2, \cdots, a_n)$ be the diagonal matrix with $a_1, a_2, \cdots, a_n$ on the diagonal. We denote $\mathcal{N}\left(0, \sigma^2\right)^{m \times n}$ as the distribution over Gaussian matrices with $m \times n$ entries, each having a standard deviation $\sigma$. For an $n \times m$ matrix $\boldsymbol{M}$, we define the *vectorization* of $\boldsymbol{M}$ as the $(n \times m)$-dimensional vector whose entries are the entries of $\boldsymbol{M}$ arranged in a sequential order.

In this paper, we assume that $G = (V, E)$ is an undirected graph with $|V| = n$ vertices, $|E| = m$ edges. For a nonempty subset $S \subset V$, we define $G[S]$ to be the induced subgraph on $S$ and we denote by $G\{S\}$ the subgraph $G[S]$, where self-loops are added to vertices $v \in S$ such that their degrees in $G$ and $G\{S\}$ are the same. For any two sets $X$ and $Y$, the symmetric difference of $X$ and $Y$ is defined as $X \triangle Y := (X \setminus Y) \cup (Y \setminus X)$. For graph $G = (V, E)$, let $\boldsymbol{D_G} := \textbf{Diag}(\mathrm{d}_G(v_1), \mathrm{d}_G(v_2), \cdots, \mathrm{d}_G(v_n))$. We denote by $\boldsymbol{A_G}$ the adjacency matrix and by $\boldsymbol{L_G}$ the Laplacian matrix where $\boldsymbol{L_G} := \boldsymbol{D_G} - \boldsymbol{A_G}$. The normalized Laplacian matrix of $G$ is defined by $\boldsymbol{\mathcal{L}_G} := \boldsymbol{D_G}^{-1/2} \boldsymbol{L_G} \boldsymbol{D_G}^{-1/2}$.

**Differential Privacy** We consider *edge-privacy* and call two graphs $G = (V, E)$ and $G' = (V', E')$ *neighboring* if it holds that $V = V'$ and $|(E \setminus E') \cup (E' \setminus E)| \leq 1$. The definition of differential privacy is as follows:

**Definition 2.1** (Differential privacy). *A randomized algorithm $\mathcal{M}$ is $(\epsilon, \delta)$-differentially private if for all neighboring graphs $G$ and $G'$ and all subsets of outputs $S$, $\Pr[\mathcal{M}(G) \in S] \leq e^{-\epsilon} \cdot \Pr[\mathcal{M}(G') \in S] + \delta$, where the probability is over the randomness of the algorithm.*

DP mechanisms typically achieve privacy by adding noise, where the magnitude of the noise depends on the sensitivity of the function.

**Definition 2.2** (Sensitivity of a function). *Let $f : \mathcal{D} \to \mathbb{R}^d$ with domain $\mathcal{D}$ be a function. The $\ell_1$-sensitivity and $\ell_2$-sensitivity of $f$ are defined respectively as*

$$\Delta_{f,1} := \max_{\substack{Y,Y' \in \mathcal{D} \\ Y,Y' \text{ are neighboring}}} \|f(Y) - f(Y')\|_1 \qquad \Delta_{f,2} := \max_{\substack{Y,Y' \in \mathcal{D} \\ Y,Y' \text{ are neighboring}}} \|f(Y) - f(Y')\|_2 .$$

Gaussian mechanism is one of the most widely used mechanisms in differential privacy.

**Lemma 2.3** (Gaussian mechanism). *Let $f : \mathcal{D} \to \mathbb{R}^d$ with domain $\mathcal{D}$ be an arbitrary $d$-dimensional function. For $0 < \epsilon, \delta \leq 1$, the algorithm that adds noise scaled to $\mathcal{N}\left(0, \frac{\Delta_{f,2}^2 \cdot 2\log(2/\delta)}{\epsilon^2}\right)$ to each of the $d$ components of $f$ is $(\epsilon, \delta)$-DP.*

**$k$-means and spectral clustering** Given a set of $n$ points $\boldsymbol{F}_1, \ldots, \boldsymbol{F}_n \in \mathbb{R}^k$, the objective of $k$-*means* problem is to find a $k$-partition of these points, $\mathcal{C} = \{C_1, C_2, \ldots, C_k\}$, such that the sum of squared distances between each data point and its assigned cluster center (i.e., the average of all points in the cluster) is minimized. This optimization problem can be formally expressed as $\arg\min_{\mathcal{C}} \sum_{i=1}^{k} \sum_{u \in C_i} \left\| u - \frac{1}{|C_i|} \sum_{x \in C_i} x \right\|_2^2$. It is known there exist polynomial time algorithms that approximate the optimum of the $k$-means within a constant factor (see e.g. Kanungo et al. (2002); Ahmadian et al. (2019)).

Peng et al. (2015) proved the approximation ratio of spectral clustering when eigenvectors satisfy certain properties. They showed the following lemma, whose proof is sketched in Appendix B.

**Lemma 2.4** (Peng et al. (2015)). *Let $G = (V, E)$ be a graph and $k \in \mathbb{N}$. Let $F : V \to \mathbb{R}^k$ be the embedding defined by $F(u) = \frac{1}{\sqrt{\mathrm{d}_G(u)}} \cdot (\boldsymbol{f_1}(u), \cdots, \boldsymbol{f_k}(u))^\top$, where $\{\boldsymbol{f_i}\}_{i=1}^k$ is a set of orthogonal bases in $\mathbb{R}^n$. Let $\{S_i\}_{i=1}^k$ be a $k$-partition of $G$, and $\{\bar{\boldsymbol{g}}_i\}_{i=1}^k$ be the normalized indicator vectors of the clusters $\{S_i\}_{i=1}^k$, where $\bar{\boldsymbol{g}}_i(u) = \sqrt{\frac{\mathrm{d}_G(u)}{\mathrm{vol}_G(S_i)}}$ if $u \in S_i$, and $\bar{\boldsymbol{g}}_i(u) = 0$ otherwise. Suppose there is a threshold $\theta \leq \frac{1}{5k}$, such that for each $i \in [k]$, there exists a linear combination of the eigenvectors $\bar{\boldsymbol{g}}_1, \cdots, \bar{\boldsymbol{g}}_k$ with coefficients $\beta_j^{(i)} : \hat{\boldsymbol{g}}_i = \beta_1^{(i)} \bar{\boldsymbol{g}}_1 + \cdots + \beta_k^{(i)} \bar{\boldsymbol{g}}_k$, and for each $i \in [k]$, $\|\boldsymbol{f_i} - \hat{\boldsymbol{g}}_i\|_2 \leq \theta$.*

*Let* KMEANS *be any algorithm for the $k$-means problem in $\mathbb{R}^k$ with approximation ratio* APT. *Let $\{A_i\}_{i=1}^k$ be a $k$-partition obtained by invoking* KMEANS *on the input set $\{F(u)\}_{u \in V}$. Then, there exists a permutation $\sigma$ on $\{1, \ldots, k\}$ such that $\mathrm{vol}_G(A_i \triangle S_{\sigma(i)}) = O(\mathrm{APT} \cdot k^2 \cdot \theta^2) \mathrm{vol}_G(S_{\sigma(i)})$ holds for every $i \in [k]$.*

## 3 STABILITY OF GENERALIZED STRONGLY CONVEX OPTIMIZATION

The following is a generalization of a result in Chen et al. (2023) whose proof is deferred to Appendix C. It shows that if the objective function of an SDP is generalized strongly convex (see Definition C.2) for some diagonal matrices $\boldsymbol{D_1}$ and $\boldsymbol{D_2}$, then the $\ell_2$-sensitivity of the scaled solution can be bounded by the $\ell_1$-sensitivity of the objective function.

**Lemma 3.1** (Stability of strongly-convex optimization). *Let $\mathcal{Y}$ be a set of databases. Let $\mathcal{K}(\mathcal{Y})$ be a family of closed convex subsets of $\mathbb{R}^m$ parameterized by $Y \in \mathcal{Y}$ and let $\mathcal{F}(\mathcal{Y})$ be a family of functions $f_Y : \mathcal{K}(Y) \to \mathbb{R}$, parameterized by $Y \in \mathcal{Y}$, such that: (1) for adjacent databases $Y, Y' \in \mathcal{Y}$, and $\boldsymbol{X} \in \mathcal{K}(Y)$ there exist $\boldsymbol{X'} \in \mathcal{K}(Y') \cap \mathcal{K}(Y)$ satisfying $|f_Y(\boldsymbol{X}) - f_{Y'}(\boldsymbol{X'})| \leq \alpha$ and $|f_{Y'}(\boldsymbol{X'}) - f_Y(\boldsymbol{X'})| \leq \alpha$. (2) $f_Y$ is $(\kappa, \boldsymbol{D_1}, \boldsymbol{D_2})$-strongly convex in $\boldsymbol{X} \in \mathcal{K}(Y)$ for some diagonal matrices $\boldsymbol{D_1}$ and $\boldsymbol{D_2}$. Then for $Y, Y' \in \mathcal{Y}$, $\widehat{\boldsymbol{X}} := \arg\min_{\boldsymbol{X} \in \mathcal{K}(Y)} f_Y(\boldsymbol{X})$ and $\widehat{\boldsymbol{X}'} := \arg\min_{\boldsymbol{X'} \in \mathcal{K}(Y')} f_{Y'}(\boldsymbol{X'})$, it holds $\left\| \boldsymbol{D_1} \widehat{\boldsymbol{X}} \boldsymbol{D_2} - \boldsymbol{D_1} \widehat{\boldsymbol{X}'} \boldsymbol{D_2} \right\|_{\mathrm{F}}^2 \leq \frac{12\alpha}{\kappa}$.*

## 4 PRIVATE CLUSTERING FOR WELL-CLUSTERED GRAPHS

In this section, we present our DP algorithm for a well-clustered graph and prove Theorem 1.

### 4.1 THE ALGORITHM

For a $c$-balanced $(k, \phi_{\text{in}}, \phi_{\text{out}})$-clusterable graph $G = (V, E)$ with a ground truth partition $\{C_i\}_{i \in [k]}$, we set $b = \frac{1}{m^2} \sum_{i,j \in [k], i \neq j} \text{vol}_G(C_i)\text{vol}_G(C_j) = 1 - \frac{1}{2m^2} \sum_{i \in [k]} \text{vol}_G(C_i)^2$. Specifically, if all clusters have the same volume, then $b = \frac{k-1}{k}$.

We make use of the following SDP (1) to extract the cluster structure of $G$. Note that the SDP assumes the knowledge of the parameter $b$, which has also been used in previous work (e.g., Guédon & Vershynin (2016)).

<div>

**SDP (1)**

$$\min \quad \sum_{(u,v) \in E} \|\bar{u} - \bar{v}\|_2^2 + \frac{2 \sum\limits_{u,v \in V} \langle \bar{u}, \bar{v} \rangle^2 \mathrm{d}_G(u)\mathrm{d}_G(v)}{\lambda m}$$

$$\text{s.t.} \quad \sum_{u,v \in V} \left( \|\bar{u} - \bar{v}\|_2^2 \mathrm{d}_G(u)\mathrm{d}_G(v) \right) \geq 2bm^2$$

$$\langle \bar{u}, \bar{v} \rangle \geq 0, \text{ for all } u, v \in V$$

$$\|\bar{u}\|_2^2 = 1, \text{ for all } u \in V$$

</div>

<div>

**SDP (2)**

$$\min \quad \langle \boldsymbol{L_G}, \boldsymbol{X} \rangle + \frac{n}{\lambda m} \|\boldsymbol{D_G^{1/2}} \boldsymbol{X} \boldsymbol{D_G^{1/2}}\|_{\mathrm{F}}^2$$

$$\text{s.t.} \quad \langle \boldsymbol{D_G L_{K_V} D_G}, \boldsymbol{X} \rangle \geq \frac{bm^2}{n}$$

$$\boldsymbol{X} \succeq 0, \boldsymbol{X} \geq 0, \boldsymbol{X}_{ii} = \frac{1}{n}, \forall i$$

</div>

Intuitively, the SDP ensures a significant sum of vector distances for all pairs of vertices, and the objective is to minimize the vector distances between endpoints of all edges, thereby achieving a configuration where inter-class vector distances are large and intra-class vector distances are small.

Let $\boldsymbol{X}$ be $\frac{1}{n}$ times the Gram matrix of these vector $\bar{\boldsymbol{v}}_1, \bar{\boldsymbol{v}}_2, \cdots, \bar{\boldsymbol{v}}_n$ (i.e., $\boldsymbol{X}_{i,j} = \frac{1}{n} \cdot \bar{\boldsymbol{v}}_i \cdot \bar{\boldsymbol{v}}_j$), $\boldsymbol{L_{K_V}}$ be the Laplacian of the complete graph on set $V$. Let $\boldsymbol{X} \geq 0$ denote that all entries of $\boldsymbol{X}$ are non-negative. It is easy to see that the SDP (1) can be equivalently written in the form of SDP (2).

Define a domain $\mathcal{D}$ as $\mathcal{D} := \left\{ \boldsymbol{X} \in \mathbb{R}^{n \times n} \,\middle|\, \langle \boldsymbol{D_G L_{K_V} D_G}, \boldsymbol{X} \rangle \geq \frac{bm^2}{n}, \boldsymbol{X} \succeq 0, \boldsymbol{X} \geq 0, \boldsymbol{X}_{ii} = \frac{1}{n}, \forall i \right\}$. Then the optimal solution of this SDP can be expressed as

$$\widehat{\boldsymbol{X}} := \arg\min_{\boldsymbol{X} \in \mathcal{D}} \langle \boldsymbol{L_G}, \boldsymbol{X} \rangle + \frac{n}{\lambda m} \|\boldsymbol{D_G^{1/2}} \boldsymbol{X} \boldsymbol{D_G^{1/2}}\|_{\mathrm{F}}^2.$$

Now we are ready to describe our algorithm whose pseudo-code is given in Algorithm 1. That is, we first solve the aforementioned SDP to obtain a solution $\boldsymbol{X_1}$ and then we add Gaussian noise to a scaled version of $\boldsymbol{X_1}$, denoted by $\boldsymbol{X_2}$. We then find the first $k$ eigenvectors of $\boldsymbol{X_2}$ and obtain the corresponding spectral embedding $\{F(u)\}_{u \in V}$ and then apply the approximation algorithm KMEANS on the embedding and output the final $k$ partition (of the vertex set $V$).

---

**Algorithm 1:** Private Clustering

**Input:** Graph $G = (V, E), \varepsilon, \delta$

**Output:** A $k$-partition $\{\widehat{C}_i\}_{i \in [k]}$

1 Let $\boldsymbol{X_1} := \arg\min_{\boldsymbol{X} \in \mathcal{D}} \langle \boldsymbol{L_G}, \boldsymbol{X} \rangle + \frac{n}{\lambda m} \|\boldsymbol{D_G^{1/2}} \boldsymbol{X} \boldsymbol{D_G^{1/2}}\|_{\mathrm{F}}^2$.

2 Let $\boldsymbol{X_2} := n\boldsymbol{D_G^{1/2}} \boldsymbol{X_1} \boldsymbol{D_G^{1/2}} + \boldsymbol{W}$, where $\boldsymbol{W} \sim \mathcal{N}\left(0, 24(\lambda + 3)m \cdot \frac{\log(2/\delta)}{\epsilon^2}\right)^{n \times n}$.

3 Let $\boldsymbol{f_1}, \boldsymbol{f_2}, ..., \boldsymbol{f_k}$ be the $k$ eigenvectors of $\boldsymbol{X_2}$ corresponding to the first $k$ smallest eigenvalues.

4 Let $F : V(G) \to \mathbb{R}^k$, where $F(u) = \mathrm{d}_G(u)^{-1/2}(\boldsymbol{f_1}(u), \boldsymbol{f_2}(u), ..., \boldsymbol{f_k}(u))^\top$.

5 Apply KMEANS($F(u), u \in V$) and let $\widehat{C}_1, \ldots, \widehat{C}_k$ be the output sets.

---

### 4.2 PROOF OF THEOREM 1

**Privacy of the algorithm**

**Lemma 4.1** (Stability). *Let* $f(G, \boldsymbol{X}) = \langle \boldsymbol{L_G}, \boldsymbol{X} \rangle + \frac{n}{\lambda m} \left\| \boldsymbol{D_G}^{1/2} \boldsymbol{X} \boldsymbol{D_G}^{1/2} \right\|_{\mathrm{F}}^2$, *and let* $g(G) = n \boldsymbol{D_G}^{1/2} \left( \arg\min_{\boldsymbol{X} \in \mathcal{D}} f(G, \boldsymbol{X}) \right) \boldsymbol{D_G}^{1/2}$. *The* $\ell_2$-*sensitivity* $\Delta_{g,2} \leq \sqrt{24(\lambda + 3)m}$.

*Proof.* For two adjacent graphs $G, G'$, we have $\|\boldsymbol{L_G} - \boldsymbol{L_{G'}}\|_{1,*} \leq 4$, where $\|\boldsymbol{L_G} - \boldsymbol{L_{G'}}\|_{1,*}$ is $\ell_1$ norm of the vectorizations of the matrix. And according to the range of $\boldsymbol{X}$ that $\boldsymbol{X} \succeq 0, \boldsymbol{X}_{ii} = \frac{1}{n}$, we have $\max_{i,j} \boldsymbol{X}_{ij} \leq \frac{1}{n}$. Thus, $|\langle \boldsymbol{L_G}, \boldsymbol{X} \rangle - \langle \boldsymbol{L_{G'}}, \boldsymbol{X} \rangle| \leq \frac{4}{n}$.

Without loss of generality, let $G'$ have one more edge $(u^*, v^*)$ than $G$. In this case,

$$\left| \left\| \boldsymbol{D_G}^{1/2} \boldsymbol{X} \boldsymbol{D_G}^{1/2} \right\|_{\mathrm{F}}^2 - \left\| \boldsymbol{D_{G'}}^{1/2} \boldsymbol{X} \boldsymbol{D_{G'}}^{1/2} \right\|_{\mathrm{F}}^2 \right| = \sum_{u,v} \boldsymbol{X}_{uv}^2 |\mathrm{d}_G(u)\mathrm{d}_G(v) - \mathrm{d}_{G'}(u)\mathrm{d}_{G'}(v)|$$

$$\leq \frac{1}{n^2} \sum_{u,v} |\mathrm{d}_G(u)\mathrm{d}_G(v) - \mathrm{d}_{G'}(u)\mathrm{d}_{G'}(v)| \leq \frac{1}{n^2} \left( 4 + 4 \sum_u \mathrm{d}_G(u) \right)$$

$$\leq \frac{8m + 4}{n^2} \leq \frac{12m}{n^2}$$

So $f$ has $\ell_1$-sensitivity $\frac{4}{n} + \frac{12m}{\lambda n m}$ with respect to $G$.

Next, we prove that $f(G, \boldsymbol{X})$ is $(\frac{2n}{\lambda m}, \boldsymbol{D_G}^{1/2}, \boldsymbol{D_G}^{1/2})$-strongly convex with respect to $\boldsymbol{X}$. The gradient $\nabla f(G, \boldsymbol{X}) = \boldsymbol{L_G} + \frac{2n}{\lambda m} \boldsymbol{D_G}^{1/2} \boldsymbol{X} \boldsymbol{D_G}^{1/2}$. Let $\boldsymbol{X}, \boldsymbol{X}' \in \mathcal{K}$ then

$$f(G, \boldsymbol{X}') = \langle \boldsymbol{L_G}, \boldsymbol{X}' \rangle + \frac{n}{\lambda m} \left\| \boldsymbol{D_G}^{1/2} \boldsymbol{X}' \boldsymbol{D_G}^{1/2} \right\|_{\mathrm{F}}^2$$

$$= \langle \boldsymbol{L_G}, \boldsymbol{X}' \rangle + \frac{n}{\lambda m} \left\| \boldsymbol{D_G}^{1/2} \boldsymbol{X}' \boldsymbol{D_G}^{1/2} - \boldsymbol{D_G}^{1/2} \boldsymbol{X} \boldsymbol{D_G}^{1/2} \right\|_{\mathrm{F}}^2$$

$$\quad - \frac{n}{\lambda m} \left\| \boldsymbol{D_G}^{1/2} \boldsymbol{X} \boldsymbol{D_G}^{1/2} \right\|_{\mathrm{F}}^2 + \frac{2n}{\lambda m} \langle \boldsymbol{D_G}^{1/2} \boldsymbol{X} \boldsymbol{D_G}^{1/2}, \boldsymbol{D_G}^{1/2} \boldsymbol{X}' \boldsymbol{D_G}^{1/2} \rangle$$

$$= \langle \boldsymbol{L_G}, \boldsymbol{X} \rangle + \langle \boldsymbol{L_G}, \boldsymbol{X}' - \boldsymbol{X} \rangle + \frac{n}{\lambda m} \left\| \boldsymbol{D_G}^{1/2} \boldsymbol{X} \boldsymbol{D_G}^{1/2} \right\|_{\mathrm{F}}^2$$

$$\quad + \frac{2n}{\lambda m} \langle \boldsymbol{D_G}^{1/2} \boldsymbol{X} \boldsymbol{D_G}^{1/2}, \boldsymbol{D_G}^{1/2} \boldsymbol{X}' \boldsymbol{D_G}^{1/2} - \boldsymbol{D_G}^{1/2} \boldsymbol{X} \boldsymbol{D_G}^{1/2} \rangle + \frac{n}{\lambda m} \left\| \boldsymbol{D_G}^{1/2} \boldsymbol{X}' \boldsymbol{D_G}^{1/2} - \boldsymbol{D_G}^{1/2} \boldsymbol{X} \boldsymbol{D_G}^{1/2} \right\|_{\mathrm{F}}^2$$

$$\geq f(G, \boldsymbol{X}) + \langle \boldsymbol{L_G}, \boldsymbol{X}' - \boldsymbol{X} \rangle + \frac{2n}{\lambda m} \langle \boldsymbol{D_G}^{1/2} \boldsymbol{X} \boldsymbol{D_G}^{1/2}, \boldsymbol{X}' - \boldsymbol{X} \rangle + \frac{n}{\lambda m} \left\| \boldsymbol{D_G}^{1/2} \boldsymbol{X}' \boldsymbol{D_G}^{1/2} - \boldsymbol{D_G}^{1/2} \boldsymbol{X} \boldsymbol{D_G}^{1/2} \right\|_{\mathrm{F}}^2$$

$$= f(G, \boldsymbol{X}) + \langle \nabla f(G, \boldsymbol{X}), \boldsymbol{X}' - \boldsymbol{X} \rangle + \frac{n}{\lambda m} \left\| \boldsymbol{D_G}^{1/2} \boldsymbol{X}' \boldsymbol{D_G}^{1/2} - \boldsymbol{D_G}^{1/2} \boldsymbol{X} \boldsymbol{D_G}^{1/2} \right\|_{\mathrm{F}}^2$$

That is $f(G, \boldsymbol{X})$ is $(\frac{2n}{\lambda m}, \boldsymbol{D_G}^{1/2}, \boldsymbol{D_G}^{1/2})$-strongly convex with respect to $\boldsymbol{X}$.

By Lemma 3.1, $\left\| \frac{g(G)}{n} - \frac{g(G')}{n} \right\|_{\mathrm{F}}^2 \leq \frac{24(\lambda+3)m}{n^2}$. So the $\ell_2$-sensitivity $\Delta_{g,2} \leq \sqrt{24(\lambda + 3)m}$. $\quad\square$

**Lemma 4.2** (Privacy). *The Algorithm 1 is* $(\epsilon, \delta)$-*DP.*

*Proof.* Consider $\boldsymbol{X_2}$ in the algorithm as a function of $G$. According to Lemma 4.1, we can get that the $\ell_2$-sensitivity of $\boldsymbol{X_2}$ is not greater than $\sqrt{24(\lambda + 3)m}$. Combining with Lemma 2.3, we get that the algorithm achieves $(\epsilon, \delta)$-DP. $\quad\square$

**Utility of the algorithm**

**Lemma 4.3** (Utility). *Suppose that* $\frac{\phi_{out}}{\phi_{in}^2} = O(c^2 k^{-4})$, $\lambda \in [\Omega(k^4 c^{-2} \phi_{in}^{-2}), O(\frac{mc^2\epsilon^2}{nk^4 \log(2/\delta)})]$. *For any* $c$-*balanced* $(k, \phi_{in}, \phi_{out})$-*clusterable graph, let* $\{\widehat{C}_i\}_{i=1}^k$ *be a* $k$-*partition obtained by the Algorithm 1 (which invokes a* $k$-*means algorithm* KMEANS *with an approximation ratio* APT*). Then, there exists a permutations* $\sigma$ *on* $\{1, \ldots, k\}$ *such that*

$$\mathrm{vol}_G(\widehat{C}_i \triangle C_{\sigma(i)}) \leq O \left( \mathrm{APT} \cdot \frac{k^4}{c^2} \cdot \left( \frac{\phi_{out} + \frac{1}{\lambda}}{\phi_{in}^2} + \lambda \frac{n}{m} \frac{\log(2/\delta)}{\epsilon^2} \right) \right) \mathrm{vol}_G(C_{\sigma(i)})$$

*with probability* $1 - \exp(-\Omega(n))$.

Note that Theorem 1 follows by choosing a constant-approximation $k$-means algorithm KMEANS (so that APT $= O(1)$) and setting $\lambda = \sqrt{\frac{m\epsilon^2}{n\log(2/\delta)}}$ and letting $m \geq n \cdot \frac{\phi_{\text{in}}^4}{\phi_{\text{out}}^2} \cdot \frac{\log(2/\delta)}{\epsilon^2}$.

To prove Lemma 4.3, we need the following lemma.

**Lemma 4.4.** *Consider the SDP (1), if the input graph $G = (V, E)$ is a $c$-balanced $(k, \phi_{in}, \phi_{out})$-clusterable graph, the solution satisfies*

$$\begin{cases} \displaystyle\sum_{u,v \in C_i} \left(\|\bar{\boldsymbol{u}} - \bar{\boldsymbol{v}}\|_2^2 \mathrm{d}_G(u)\mathrm{d}_G(v)\right) \leq m\mathrm{vol}_G(C_i) \cdot \frac{\phi_{out} + \frac{16}{\lambda}}{\phi_{in}^2}, \quad \forall i \in [k] \\ \displaystyle\sum_{u \in C_i, v \in C_j, i \neq j} \left(\|\bar{\boldsymbol{u}} - \bar{\boldsymbol{v}}\|_2^2 \mathrm{d}_G(u)\mathrm{d}_G(v)\right) \geq m^2 \left(2b - \frac{\phi_{out} + \frac{16}{\lambda}}{\phi_{in}^2}\right). \end{cases}$$

*Proof.* As $G = (V, E)$ is a $c$-balanced $(k, \phi_{\text{in}}, \phi_{\text{out}})$-clusterable graph, there exists a $c$-balanced partitioning $\{C_i\}_{i \in [k]}$ of $V$ and, for all $i \in [k]$, $\Phi_{\text{in}}(G, C_i) \geq \phi_{\text{in}}$ and $\Phi_{\text{out}}(G, C_i) \leq \phi_{\text{out}}$.

For $1 \leq i \leq k$, we have $\Phi_{\text{out}}(G, C_i) = \frac{|E(C_i, V - C_i)|}{\mathrm{vol}_G(C_i)} \leq \phi_{\text{out}}$. Summing it yields $\sum_{i=1}^{k} |E(C_i, V - C_i)| \leq \phi_{\text{out}} \cdot \sum_{i=1}^{k} \mathrm{vol}_G(C_i) = \phi_{\text{out}} \cdot \mathrm{vol}_G(V) = 2\phi_{\text{out}}m$. So the number of edges between clusters in $G$ is not greater than $\phi_{\text{out}}m$.

Now let us consider a feasible solution of the SDP (1). For every vertex $u$ in the $\ell$-th cluster, assign unit vector $\boldsymbol{v_\ell}$ to $\bar{\boldsymbol{u}}$. We can let $\boldsymbol{v_1}, \boldsymbol{v_2}, \ldots, \boldsymbol{v_k}$ be a set of orthogonal bases, as $k \leq n$. For this feasible solution, the value of the objective function is not greater than $\phi_{\text{out}}m \cdot 2 + \frac{2}{\lambda m} \cdot \sum_{j \in [k]} \mathrm{vol}_G(C_j)^2 \leq 2\phi_{\text{out}}m + \frac{2}{\lambda m} \cdot \mathrm{vol}_G(G)^2 = 2\phi_{\text{out}}m + \frac{8m}{\lambda}$.

Thus, for any $i \in [k]$, we have $\sum_{(u,v) \in G\{C_i\}} \|\bar{\boldsymbol{u}} - \bar{\boldsymbol{v}}\|_2^2 \leq 2\phi_{\text{out}}m + \frac{8m}{\lambda}$. Let $\mu$ be the second eigenvalue of the normalized Laplacian matrix $\boldsymbol{\mathcal{L}_{G\{C_i\}}}$, we have

$$\mu = \mathrm{vol}_G(C_i) \min_{\{\bar{\boldsymbol{u}}\}u \in V} \frac{\sum_{(u,v) \in G\{C_i\}} \left(\|\bar{\boldsymbol{u}} - \bar{\boldsymbol{v}}\|_2^2\right)}{\sum_{u,v \in C_i} \left(\|\bar{\boldsymbol{u}} - \bar{\boldsymbol{v}}\|_2^2\right) \mathrm{d}_G(u)\mathrm{d}_G(v)} \geq \frac{\phi_{\text{in}}^2}{2}$$

So $\sum_{u,v \in C_i} \left(\|\bar{\boldsymbol{u}} - \bar{\boldsymbol{v}}\|_2^2 \mathrm{d}_G(u)\mathrm{d}_G(v)\right) \leq \mathrm{vol}_G(C_i) \cdot \frac{2\phi_{\text{out}}m + \frac{8m}{\lambda}}{\frac{1}{2}\phi_{\text{in}}^2} = \mathrm{vol}_G(C_i) \cdot \frac{4\phi_{\text{out}}m + \frac{16m}{\lambda}}{\phi_{\text{in}}^2} = m\mathrm{vol}_G(C_i) \cdot \frac{4\phi_{\text{out}} + \frac{16}{\lambda}}{\phi_{\text{in}}^2}$, for all $i \in [k]$.

By the definition of the SDP (1), we know that $\sum_{u,v \in V} \left(\|\bar{\boldsymbol{u}} - \bar{\boldsymbol{v}}\|_2^2 \mathrm{d}_G(u)\mathrm{d}_G(v)\right) \geq 2bm^2$.

Thus, $\sum_{u \in C_i, v \in C_j, i \neq j} \left(\|\bar{\boldsymbol{u}} - \bar{\boldsymbol{v}}\|_2^2 \mathrm{d}_G(u)\mathrm{d}_G(v)\right) \geq 2bm^2 - \sum_{i \in [k]} \left(m\mathrm{vol}_G(C_i) \cdot \frac{4\phi_{\text{out}} + \frac{16}{\lambda}}{\phi_{\text{in}}^2}\right) \geq 2bm^2 - m \cdot \frac{4\phi_{\text{out}} + \frac{16}{\lambda}}{\phi_{\text{in}}^2} \sum_{i \in [k]} \mathrm{vol}_G(C_i) = m^2 \left(2b - \frac{4\phi_{\text{out}} + \frac{16}{\lambda}}{\phi_{\text{in}}^2}\right)$. $\square$

**Lemma 4.5.** *Consider the setting in Theorem 1 and Algorithm 1, with probability $1 - \exp(-\Omega(n))$,*

$$\|\boldsymbol{X_2} - \boldsymbol{Z}\|_2 \leq 2m \cdot \left(2\sqrt{\frac{\phi_{out} + \frac{4}{\lambda}}{\phi_{in}^2}} + 3\sqrt{\frac{6(\lambda + 3)n}{m} \frac{\log(2/\delta)}{\epsilon^2}}\right)$$

*where $\boldsymbol{Z}$ is a matrix defined as: $\boldsymbol{Z}_{uv} = \sqrt{\mathrm{d}_G(u)\mathrm{d}_G(v)}$, if $u, v$ are in same cluster; $\boldsymbol{Z}_{uv} = 0$ if $u, v$ are in different clusters.*

*Proof.* According to Lemma 4.4, for $u, v$ in same cluster $C_i$ :

$$\sum_{u,v \in C_i} \left(n\sqrt{\mathrm{d}_G(u)\mathrm{d}_G(v)}[\boldsymbol{X_1}]_{uv} - \boldsymbol{Z}_{uv}\right)^2 = \sum_{u,v \in C_i} \left((n[\boldsymbol{X_1}]_{uv} - 1)^2 \mathrm{d}_G(u)\mathrm{d}_G(v)\right)$$

$$\leq 2 \sum_{u,v \in C_i} (|n[\boldsymbol{X_1}]_{uv} - 1|\,\mathrm{d}_G(u)\mathrm{d}_G(v)) = \sum_{u,v \in C_i} \left(\|\bar{\boldsymbol{u}} - \bar{\boldsymbol{v}}\|_2^2 \mathrm{d}_G(u)\mathrm{d}_G(v)\right)$$

$$\leq m\mathrm{vol}_G(C_i) \cdot \frac{4\phi_{\text{out}} + \frac{16}{\lambda}}{\phi_{\text{in}}^2}$$

For $u, v$ in different clusters :

$$\sum_{\substack{u\in C_i, v\in C_j \\ i\neq j}} \left(n\sqrt{\mathrm{d}_G(u)\mathrm{d}_G(v)}[\boldsymbol{X_1}]_{uv} - \boldsymbol{Z}_{uv}\right)^2 = \sum_{\substack{u\in C_i, v\in C_j \\ i\neq j}} \left((n[\boldsymbol{X_1}]_{uv})^2\,\mathrm{d}_G(u)\mathrm{d}_G(v)\right)$$

$$\leq \sum_{\substack{u\in C_i, v\in C_j \\ i\neq j}} \left(|n[\boldsymbol{X_1}]_{uv}|\,\mathrm{d}_G(u)\mathrm{d}_G(v)\right) = \frac{1}{2}\sum_{\substack{u\in C_i, v\in C_j \\ i\neq j}} \left(2 - \|\bar{\boldsymbol{u}}-\bar{\boldsymbol{v}}\|_2^2\right)\mathrm{d}_G(u)\mathrm{d}_G(v)$$

$$= bm^2 - \frac{1}{2}\sum_{\substack{u\in C_i, v\in C_j \\ i\neq j}} \left(\|\bar{\boldsymbol{u}}-\bar{\boldsymbol{v}}\|_2^2\mathrm{d}_G(u)\mathrm{d}_G(v)\right) \leq m^2\cdot\frac{2\phi_{\mathrm{out}}+\frac{8}{\lambda}}{\phi_{\mathrm{in}}^2}$$

Combing these, we have: $\left\|n\boldsymbol{D_G}^{1/2}\boldsymbol{X_1}\boldsymbol{D_G}^{1/2} - \boldsymbol{Z}\right\|_2 \leq \left\|n\boldsymbol{D_G}^{1/2}\boldsymbol{X_1}\boldsymbol{D_G}^{1/2} - \boldsymbol{Z}\right\|_{\mathrm{F}}$
$= \sqrt{\sum_{u,v}\left(n\sqrt{\mathrm{d}_G(u)\mathrm{d}_G(v)}[\boldsymbol{X_1}]_{uv} - \boldsymbol{Z}_{uv}\right)^2} \leq 4m\cdot\sqrt{\frac{\phi_{\mathrm{out}}+\frac{4}{\lambda}}{\phi_{\mathrm{in}}^2}}.$

Algorithm 1 uses $\boldsymbol{W}\sim\mathcal{N}\left(0, 24\left(\lambda+3\right)m\cdot\frac{\log(2/\delta)}{\epsilon^2}\right)^{n\times n}$, and we choose $t=\sqrt{n}$ in Lemma A.1, so with probability $1 - \exp(-\Omega(n))$, $\|\boldsymbol{X_2}-\boldsymbol{Z}\|_2 \leq \|\boldsymbol{W}\|_2 + \left\|n\boldsymbol{D_G}^{1/2}\boldsymbol{X_1}\boldsymbol{D_G}^{1/2} - \boldsymbol{Z}\right\|_2 \leq 3\sqrt{n}\cdot$
$\sqrt{24\left(\lambda+3\right)m\cdot\frac{\log(2/\delta)}{\epsilon^2}} + 4m\cdot\sqrt{\frac{\phi_{\mathrm{out}}+\frac{4}{\lambda}}{\phi_{\mathrm{in}}^2}} = 2m\cdot\left(2\sqrt{\frac{\phi_{\mathrm{out}}+\frac{4}{\lambda}}{\phi_{\mathrm{in}}^2}} + 3\sqrt{\frac{6(\lambda+3)n}{m}\frac{\log(2/\delta)}{\epsilon^2}}\right).$ $\qquad\square$

*Proof of Lemma 4.3.* Let $\gamma = 2\sqrt{\frac{\phi_{\mathrm{out}}+\frac{4}{\lambda}}{\phi_{\mathrm{in}}^2}} + 3\sqrt{\frac{6(\lambda+3)n}{m}\frac{\log(2/\delta)}{\epsilon^2}}$. According to Lemma 4.5, with probability $1 - \exp(-\Omega(n))$, it holds that $\|\boldsymbol{X_2}-\boldsymbol{Z}\|_2 \leq 2\gamma m$.

We denote the eigenvalues of the matrix $\boldsymbol{X_2}$ by $\mu_1 \geq \cdots \geq \mu_n$, with their corresponding orthonormal eigenvectors $\boldsymbol{f_1}, \cdots, \boldsymbol{f_n}$. We denote the eigenvalues of the matrix $\boldsymbol{Z}$ by $\nu_1 \geq \cdots \geq \nu_n$, with their corresponding orthonormal eigenvectors $\boldsymbol{g_1}, \cdots, \boldsymbol{g_n}$. Let $\boldsymbol{Y} = [\boldsymbol{f_1}, \cdots, \boldsymbol{f_n}], \boldsymbol{Q} = [\boldsymbol{g_1}, \cdots, \boldsymbol{g_n}]$, and let $\boldsymbol{A} = \mathbf{Diag}(\mu_1, \cdots, \mu_n), \boldsymbol{\Lambda} = \mathbf{Diag}(\nu_1, \cdots, \nu_n)$. Then $\boldsymbol{X_2} = \boldsymbol{YAY}^\top, \boldsymbol{Z} = \boldsymbol{Q\Lambda Q}^\top$ are the eigen-decompositions of $\boldsymbol{X_2}, \boldsymbol{Z}$, respectively. As $\{\boldsymbol{g_1}, \cdots, \boldsymbol{g_n}\}$ is a set of orthogonal bases in $\mathbb{R}^n$, for every $i \in [n]$, $\boldsymbol{f_i}$ is the linear combination of eigenvector $\boldsymbol{g_1}, \cdots, \boldsymbol{g_n}$. We write $\boldsymbol{f_i}$ as $\boldsymbol{f_i} = \beta_1^{(i)}\boldsymbol{g_1} + \cdots + \beta_n^{(i)}\boldsymbol{g_n}$, where $\beta_j^{(i)} \in \mathbb{R}$. By the definition of $\boldsymbol{Z}$, we know that $\boldsymbol{Z}$ is composed of $k$ rank-1 matrices of sizes $|C_1|, |C_2|, \cdots, |C_k|$, respectively. Let these matrices be $\boldsymbol{M_1}, \boldsymbol{M_2}, \cdots, \boldsymbol{M_k}$ such that $\boldsymbol{M_i} \in \mathbb{R}^{|C_i|\times|C_i|}$, for each $i \in [k]$. For each $j \in [k]$, note that the eigenvalues of $\boldsymbol{M_j}$ are $\mathrm{vol}_G(C_j), 0, \cdots, 0$, where the multiplicities of $0$ is $|C_j| - 1$. So we have $\nu_{k+1} = \cdots = \nu_n = 0$, and $\nu_1, \cdots, \nu_k$ are equal to $\mathrm{vol}_G(C_1), \cdots, \mathrm{vol}_G(C_k)$, respectively, and $\nu_k = \min_{i\in[k]}\mathrm{vol}_G(C_i) \geq \frac{2cm}{k}$.

By Lemma A.2, we have $\mu_k \geq \nu_k - \|\boldsymbol{X_2}-\boldsymbol{Z}\|_2 \geq 2m\left(\frac{c}{k}-\gamma\right)$.

We apply Theorem A.3 with $H = \boldsymbol{X_2}, \boldsymbol{E_0} = \boldsymbol{Y}_{[k]}, \boldsymbol{E_1} = \boldsymbol{Y}_{-[k]}, \boldsymbol{A_0} = \boldsymbol{A}_{[k]}, \boldsymbol{A_1} = \boldsymbol{A}_{-[k]}$, and $\widetilde{H} = \boldsymbol{Z}, \boldsymbol{F_0} = \boldsymbol{Q}_{[k]}, \boldsymbol{F_1} = \boldsymbol{Q}_{-[k]}, \boldsymbol{\Lambda_0} = \boldsymbol{\Lambda}_{[k]}, \boldsymbol{\Lambda_1} = \boldsymbol{\Lambda}_{-[k]}, \eta = |\mu_k - \nu_{k+1}| = |\mu_k| \geq 2m\left(\frac{c}{k}-\gamma\right)$. Therefore, by Theorem A.3 we have

$$\left\|\boldsymbol{Q}_{-[k]}^\top\boldsymbol{Y}_{[k]}\right\|_2 = \left\|\boldsymbol{F_1}^\top\boldsymbol{E_0}\right\|_2 \leq \frac{\left\|\boldsymbol{F_1}^\top(\boldsymbol{Z}-\boldsymbol{X_2})\boldsymbol{E_0}\right\|_2}{\eta} \leq \frac{\|\boldsymbol{X_2}-\boldsymbol{Z}\|_2}{\eta} \leq \frac{\gamma k}{c-\gamma k}.$$

Thus we have $\sum_{j=k+1}^n\left(\beta_j^{(i)}\right)^2 = \left\|\boldsymbol{Q}_{-[k]}^\top\boldsymbol{f_i}\right\|_2^2 \leq \left\|\boldsymbol{Q}_{-[k]}^\top\boldsymbol{Y}_{[k]}\right\|_2^2 \leq \frac{\gamma^2 k^2}{(c-\gamma k)^2}$, for all $i \in [k]$.

For all $i \in [k]$, let $\widehat{\boldsymbol{g_i}} = \beta_1^{(i)}\boldsymbol{g_1} + \cdots + \beta_k^{(i)}\boldsymbol{g_n}$, then $\|\boldsymbol{f_i}-\widehat{\boldsymbol{g_i}}\|_2^2 = \sum_{j=k+1}^n\left(\beta_j^{(i)}\right)^2 \leq \frac{\gamma^2 k^2}{(c-\gamma k)^2}$.

Note that our Algorithm 1 invokes KMEANS algorithm on the input set the input set $\{F(u)\}_{u\in V}$, where $F(u) = \mathrm{d}_G(u)^{-1/2}(\boldsymbol{f_1}(u), \boldsymbol{f_2}(u), ..., \boldsymbol{f_k}(u))^\top$. By Lemma 2.4, we know that the

output partition $\{\widehat{C}_i\}_{i\in[k]}$ satisfies: $\mathrm{vol}_G(\widehat{C}_i\triangle C_{\sigma(i)}) \leq \mathrm{APT} \cdot k^2\frac{\gamma^2 k^2}{(c-\gamma k)^2}\mathrm{vol}_G(C_{\sigma(i)}) = O\left(\mathrm{APT}\cdot\frac{k^4}{c^2}\cdot\left(\frac{\phi_{\mathrm{out}}+\frac{1}{\lambda}}{\phi_{\mathrm{in}}^2}+\lambda\frac{n}{m}\frac{\log(2/\delta)}{\epsilon^2}\right)\right)\mathrm{vol}_G(C_{\sigma(i)})$, for some permutation $\sigma$ on $\{1,\ldots,k\}$. $\qquad\square$

## 5 EXPERIMENTS

To evaluate the empirical trade-off between privacy and utility of our algorithm, we perform exemplary experiments on synthetic datasets sampled from SBMs. As a baseline, we compare our algorithm to an approach based on randomized response as described in Mohamed et al. (2022). The algorithm based on randomized response generates a noisy output by flipping each bit of the adjacency matrix with some probability $p_{RR}$ (for undirected graphs, it flips a single coin for opposing directed edges). It was shown that randomized response is $\epsilon$-DP for $p_{RR}\geq 1/(1+e^\epsilon)$.

Since randomized response is an $\epsilon$-DP algorithm, we are interested in the utility improvement that we can gain by using $(\epsilon,\delta)$-DP when the utility of randomized response becomes insufficient. Our hypothesis is that when the noise added to the adjacency matrix by randomized response has roughly the same magnitude as the clustering signal, the output of Algorithm 1 still has significant utility. In particular, we consider the case where the difference between the empirical probability of intra-cluster edges and inter-cluster edges after randomized response is only a constant fraction of the original difference between these probabilities in the vanilla SBM graph.

**Implementation.** We model the SDP (2) in CVXPY 1.3.2 and use SCS 3.2.3 as SDP solver. We use NumPy 1.23.5 for numerical computation and scikit-learn 1.3 for an implementation of k-means++. For our algorithm, we use $b=(k-1)/k$ and $\lambda=c\cdot\sqrt{\frac{m\epsilon^2}{n\log(2/\delta)}}$, where $c$ is a trade-off constant that we fix for each SBM parameterization before sampling the input datasets. For randomized response, we replace the objective with $\arg\min_{X\in\mathcal{D}}\langle L_G, X\rangle$, i.e., we remove the regularizer term, and set $p_{RR}=1/(1+e^\epsilon)$. The regularizer facilitates a trivial solution and is not needed because randomized response is differentially private.

**Datasets and setup.** We sample datasets from a stochastic block model $\mathrm{SBM}(n,k,p,q)$ with $k$ blocks, each of size $n/k$, intra-cluster edge probability $p$ and inter-cluster edge probability $q$. We consider the sampled graphs as instances of well-clustered graphs. For our experiments, we use $\epsilon=1$ and $\delta=1/n^2$. Since $p_{RR}=1/(1+e)\approx 0.27$, we choose small values of $p,q$ so that $|p-q|=0.2$. For each parameterization, we sample 10 SBM graphs and run each algorithm 100 times to boost the statistical stability of the evaluation. See appendix E for more details.

| Parameters | | | | | RR+SDP | | Algorithm 1 | |
|---|---|---|---|---|---|---|---|---|
| n | k | p | q | c | AMI | NMI | AMI | NMI |
| 100 | 2 | 0.20 | 0.00 | 5e-6 | 0.10 | 0.11 | **0.17** | 0.19 |
| 100 | 2 | 0.25 | 0.05 | 3.5e-6 | 0.10 | 0.11 | **0.14** | 0.15 |
| 100 | 2 | 0.30 | 0.10 | 2e-6 | 0.09 | 0.10 | **0.26** | 0.27 |
| 150 | 3 | 0.20 | 0.00 | 3e-6 | 0.07 | 0.08 | **0.19** | 0.20 |
| 150 | 3 | 0.25 | 0.05 | 8e-7 | 0.06 | 0.06 | **0.57** | 0.58 |
| 150 | 3 | 0.30 | 0.10 | 7e-7 | 0.06 | 0.07 | **0.35** | 0.55 |

**Table 1:** Adjusted mutual information (AMI) and V-measure / normalized mutual information (NMI) of RR+SDP and our algorithm. Reported numbers are median values over 100 runs on the same graph, over 10 different graphs sampled from $SBM(n,k,p,q)$.

**Evaluation.** The evaluation of the comparison between RR+SDP and Algorithm 1 is shown in Table 1. We report the median over the repetitions, over the datasets, of the adjusted (AMI) and the normalized mutual information (NMI) between the ground truth clusters of the SBM model and the clustering reported by the algorithm as a measure of the mutual dependence between these two (larger values are better). From the results, we see that the noisy adjacency matrix output by randomized response obfuscates most of the signal from the original adjacency matrix so that the solution of the SDP has low utility. On the other hand, we see that the regularizer term in the SDP and the noise added to the solution recovered significantly more information from the ground truth: Using Algorithm 1 instead of randomized response can lead to $(\epsilon,\delta)$-differentially private solution with significantly improved quality, and consistently did so in our experiments.

ACKNOWLEDGMENTS

W.H. and P.P. are supported in part by NSFC grant 62272431 and "the Fundamental Research Funds for the Central Universities".

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

## A  USEFUL TOOLS

The spectral norm of a Gaussian matrix has a high probability upper bound. The following lemma illustrates this fact.

**Lemma A.1** (Concentration of spectral norm of Gaussian matrices). *Let $W \sim \mathcal{N}(0,1)^{m \times n}$. Then for any $t$, we have*

$$\Pr\left(\sqrt{m} - \sqrt{n} - t \leq \sigma_{\min}(W) \leq \sigma_{\max}(W) \leq \sqrt{m} + \sqrt{n} + t\right) \geq 1 - 2\exp\left(-\frac{t^2}{2}\right),$$

*where $\sigma_{\min}(\cdot)$ and $\sigma_{\max}(\cdot)$ denote the minimum and the maximum singular values of a matrix, respectively.*

*Let $W'$ be an $n$-by-$n$ symmetric matrix with independent entries sampled from $\mathcal{N}(0, \sigma^2)$. Then by the above fact, $\|W'\|_2 \leq 3\sigma\sqrt{n}$ with probability at least $1 - \exp(-\Omega(n))$.*

When a matrix undergoes a slight perturbation under some conditions, its eigenvalues and eigenvectors do not experience significant changes. Lemma A.2 and Lemma A.3 illustrate this.

**Lemma A.2** (Weyl's inequality). *Let $\boldsymbol{A}$ and $\boldsymbol{B}$ be symmetric matrices. Let $\boldsymbol{R} = \boldsymbol{A} - \boldsymbol{B}$. Let $\alpha_1 \geq \cdots \geq \alpha_n$ be the eigenvalues of $\boldsymbol{A}$. Let $\beta_1 \geq \cdots \geq \beta_n$ be the eigenvalues of $\boldsymbol{B}$. Then for each $i \in [n]$,*

$$|\alpha_i - \beta_i| \leq \|\boldsymbol{R}\|_2.$$

**Lemma A.3** (Davis-Kahan $\sin(\theta)$-Theorem (Davis & Kahan, 1970)). *Let $\boldsymbol{H} = \boldsymbol{E_0}\boldsymbol{A_0}\boldsymbol{E_0}^\top + \boldsymbol{E_1}\boldsymbol{A_1}\boldsymbol{E_1}^\top$ and $\widetilde{\boldsymbol{H}} = \boldsymbol{F_0}\boldsymbol{\Lambda_0}\boldsymbol{F_0}^\top + \boldsymbol{F_1}\boldsymbol{\Lambda_1}\boldsymbol{F_1}^\top$ be symmetric real-valued matrices with $\boldsymbol{E_0}, \boldsymbol{E_1}$ and $\boldsymbol{F_0}, \boldsymbol{F_1}$ orthogonal. If the eigenvalues of $\boldsymbol{A_0}$ are contained in an interval $(a, b)$, and the eigenvalues of $\boldsymbol{\Lambda_1}$ are excluded from the interval $(a - \eta, b + \eta)$ for some $\eta > 0$, then for any unitarily invariant norm $\|.\|$*

$$\|\boldsymbol{F_1}^\top \boldsymbol{E_0}\| \leq \frac{\|\boldsymbol{F_1}^\top (\widetilde{\boldsymbol{H}} - \boldsymbol{H})\boldsymbol{E_0}\|}{\eta}.$$

# B    DEFERRED PROOFS FROM SECTION 2

In this section, we give a sketch of the proof of Lemma 2.4 restarted below.

**Lemma B.1** (Peng et al. (2015)). *Let $G = (V, E)$ be a graph and $k \in \mathbb{N}$. Let $F : V \to \mathbb{R}^k$ be the embedding defined by $F(u) = \frac{1}{\sqrt{\mathrm{d}_G(u)}} \cdot (\boldsymbol{f_1}(u), \cdots, \boldsymbol{f_k}(u))^\top$, where $\{\boldsymbol{f_i}\}_{i=1}^k$ is a set of orthogonal bases in $\mathbb{R}^n$. Let $\{S_i\}_{i=1}^k$ be a $k$-partition of $G$, and $\{\bar{\boldsymbol{g}}_i\}_{i=1}^k$ be the normalized indicator vectors of the clusters $\{S_i\}_{i=1}^k$, where $\bar{\boldsymbol{g}}_i(u) = \sqrt{\frac{\mathrm{d}_G(u)}{\mathrm{vol}_G(S_i)}}$ if $u \in S_i$, and $\bar{\boldsymbol{g}}_i(u) = 0$ otherwise. Suppose there is a threshold $\theta \leq \frac{1}{5k}$, such that for each $i \in [k]$, there exists a linear combination of the eigenvectors $\bar{\boldsymbol{g}}_1, \cdots, \bar{\boldsymbol{g}}_k$ with coefficients $\beta_j^{(i)} : \widehat{\boldsymbol{g}}_i = \beta_1^{(i)}\bar{\boldsymbol{g}}_1 + \cdots + \beta_k^{(i)}\bar{\boldsymbol{g}}_k$, and for each $i \in [k]$, $\|\boldsymbol{f_i} - \widehat{\boldsymbol{g}}_i\|_2 \leq \theta$.*

*Let* KMEANS *be any algorithm for the $k$-means problem in $\mathbb{R}^k$ with approximation ratio* APT. *Let $\{A_i\}_{i=1}^k$ be a $k$-partition obtained by invoking* KMEANS *on the input set $\{F(u)\}_{u \in V}$. Then, there exists a permutation $\sigma$ on $\{1, \ldots, k\}$ such that $\mathrm{vol}_G(A_i \triangle S_{\sigma(i)}) = O(\mathrm{APT} \cdot k^2 \cdot \theta^2)\mathrm{vol}_G(S_{\sigma(i)})$ holds for every $i \in [k]$.*

*Proof sketch.* The following five properties given in Lemma B.2 are five key components proven in Peng et al. (2015). They start by demonstrating the first property based on the existing conditions. Then, they provide $k$ centers for spectral embeddings and sequentially prove that all embedded points concentrate around their corresponding centers, the magnitudes of center vectors, and the distances. Finally, based on these properties, they establish the fifth one. With this fifth property, we can directly employ a proof by contradiction to derive our conclusion. □

**Lemma B.2.** *Consider the setting in Lemma 2.4, let $\zeta \triangleq \frac{1}{10\sqrt{k}}$, $\boldsymbol{p^{(i)}} \triangleq \frac{1}{\sqrt{\mathrm{vol}(S_i)}} \left(\beta_i^{(1)}, \ldots, \beta_i^{(k)}\right)^\top$ for $i \in [k]$, and $\mathrm{COST}(C_1, \cdots, C_k) \triangleq \min_{c_1, \cdots, c_k \in \mathbb{R}^k} \sum_{i=1}^k \sum_{u \in C_i} \mathrm{d}(u) \|F(u) - c_i\|_2^2$ for the partition $C_1, \cdots, C_k$. The following statements hold:*

1. *For any $\ell \neq j$, there exists $i \in [k]$ such that*

$$\left|\beta_\ell^{(i)} - \beta_j^{(i)}\right| \geq \zeta \triangleq \frac{1}{10\sqrt{k}}$$

2. *All embedded points are concentrated around $\boldsymbol{p^{(i)}}$:*

$$\sum_{i=1}^k \sum_{u \in S_i} \mathrm{d}(u) \left\|F(u) - \boldsymbol{p^{(i)}}\right\|_2^2 \leq k\theta^2.$$

3. *For every $i \in [k]$ that*

$$\frac{99}{100\mathrm{vol}(S_i)} \leq \left\|\boldsymbol{p^{(i)}}\right\|_2^2 \leq \frac{101}{100\mathrm{vol}(S_i)}.$$

4. *For every $i \neq j, i \in [k]$, it holds that*

$$\left\| \boldsymbol{p}^{(i)} - \boldsymbol{p}^{(j)} \right\|_2^2 \geq \frac{\zeta^2}{10 \min \left\{ \operatorname{vol}(S_i), \operatorname{vol}(S_j) \right\}},$$

5. *Suppose that, for every permutations $\sigma$ on $\{1, \ldots, k\}$, there exists $i$ such that $\operatorname{vol}\left(A_i \triangle S_{\sigma(i)}\right) \geq 2\epsilon \operatorname{vol}\left(S_{\sigma(i)}\right)$ for $\epsilon \geq 10^5 \cdot k^2 \theta^2$, then $\operatorname{COST}(A_1, \ldots, A_k) \geq 10^{-4} \cdot \epsilon / k$.*

## C  DEFERRED PROOFS FROM SECTION 3

**Lemma C.1** (See e.g. (Chen et al., 2023)). *Let $f : \mathbb{R}^m \to \mathbb{R}$ be a convex function. Let $\mathcal{K} \subseteq \mathbb{R}^m$ be a convex set. Then $y^* \in \mathcal{K}$ is a minimizer of $f$ over $\mathcal{K}$ if and only if there exists a subgradient $g \in \partial f(y^*)$ such that*

$$\langle y - y^*, g \rangle \geq 0 \quad \forall y \in \mathcal{K}.$$

Since the above lemma is stated as a property of functions of vectors, in the following, we will treat a matrix as its respective vectorization. Specifically, the vectorization of a $m \times n$ matrix $\boldsymbol{A}$ is the $mn \times 1$ column vector obtained by stacking the columns of the matrix $\boldsymbol{A}$ on top of one another. The inner product of matrices $\boldsymbol{A}$ and $\boldsymbol{B}$, as well as the $\ell_2$ norm of matrix $\boldsymbol{A}$, are defined as $\langle \boldsymbol{A}, \boldsymbol{B} \rangle := \sum_{i,j} \boldsymbol{A}_{ij} \boldsymbol{B}_{ij}$ and $\|\boldsymbol{A}\|_{2,*} := \sqrt{\sum_{i,j} \boldsymbol{A}_{ij}^2}$, which are the same as the respective inner product and $\ell_2$ norm of their vectorizations. Notice that $\|\boldsymbol{A}\|_{2,*} = \|\boldsymbol{A}\|_F$, which is the Frobenius norm of the matrix $\boldsymbol{A}$.

**Definition C.2** (Generalized strongly convex function). *Let $\mathcal{K} \subseteq \mathbb{R}^{m \times n}$ be a convex set, $f : \mathcal{K} \to \mathbb{R}$ be a function, and $\boldsymbol{D_1}, \boldsymbol{D_2}$ be diagonal matrices. The function $f$ is called $(\kappa, \boldsymbol{D_1}, \boldsymbol{D_2})$-strongly convex if the following inequality holds for all $\boldsymbol{X}, \boldsymbol{X'} \in \mathcal{K}$:*

$$f(\boldsymbol{X'}) \geq f(\boldsymbol{X}) + \langle \boldsymbol{X'} - \boldsymbol{X}, \nabla f(\boldsymbol{X}) \rangle + \frac{\kappa}{2} \|\boldsymbol{D_1} \boldsymbol{X'} \boldsymbol{D_2} - \boldsymbol{D_1} \boldsymbol{X} \boldsymbol{D_2}\|_{2,*}^2$$

**Lemma C.3** (Pythagorean theorem from strong convexity). *Let $\mathcal{K} \subseteq \mathbb{R}^{m \times n}$ be a convex set, $f : \mathcal{K} \to \mathbb{R}$ be a function. Suppose $f$ is $(\kappa, \boldsymbol{D_1}, \boldsymbol{D_2})$-strongly convex for some diagonal matrices $\boldsymbol{D_1}$ and $\boldsymbol{D_2}$. Let $\boldsymbol{X}^* \in \mathcal{K}$ be a minimizer of $f$. Then for any $\boldsymbol{X} \in \mathcal{K}$, one has*

$$\|\boldsymbol{D_1} \boldsymbol{X} \boldsymbol{D_2} - \boldsymbol{D_1} \boldsymbol{X}^* \boldsymbol{D_2}\|_{2,*}^2 \leq \frac{2}{\kappa} (f(\boldsymbol{X}) - f(\boldsymbol{X}^*)).$$

*Proof.* By the definition of $(\kappa, \boldsymbol{D_1}, \boldsymbol{D_2})$-strongly convexity, for any subgradient $g \in \partial f(\boldsymbol{X}^*)$,

$$f(\boldsymbol{X}) \geq f(\boldsymbol{X}^*) + \langle \boldsymbol{X} - \boldsymbol{X}^*, g \rangle + \frac{\kappa}{2} \|\boldsymbol{D_1} \boldsymbol{X} \boldsymbol{D_2} - \boldsymbol{D_1} \boldsymbol{X}^* \boldsymbol{D_2}\|_{2,*}^2.$$

By Lemma C.1, there exists a subgradient $g \in \partial f(\boldsymbol{X}^*)$ such that $\langle \boldsymbol{X} - \boldsymbol{X}^*, g \rangle \geq 0$. Then the result follows. $\square$

*Proof of Lemma 3.1.* The proof follows from Lemma C.3 and the proof of Lemma 4.1 in (Chen et al., 2023). $\square$

## D  LOWER BOUND

In this section, we show that approximation algorithms for recovering the clusters of well-clustered, sparse graphs cannot satisfy pure $\epsilon$-DP for small error. Given a cluster membership vector $\boldsymbol{u} \in \{-1, 1\}^n$ of a graph $G = (V, E)$ that assigns each vertex to one of two clusters (labeled $-1$ and $1$), and given a ground truth vector $\boldsymbol{u_G} \in \{-1, 1\}^n$, we define the misclassification rate $\operatorname{err}(\boldsymbol{u}, \boldsymbol{u_G}) = \operatorname{err}_G(\boldsymbol{u}) = (n - \min(\langle \boldsymbol{u}, \boldsymbol{u_G} \rangle, \langle -\boldsymbol{u}, \boldsymbol{u_G} \rangle)) / (2n)$. In other words, the misclassification rate is the minimum number of assignment changes that are required to turn $\boldsymbol{u_G}$ into one of $\boldsymbol{u}$ and $-\boldsymbol{u}$. It is known that $\operatorname{err}$ is a semimetric.

**Lemma D.1** (Lemma 5.25, Chen et al. (2023)). *For any $\boldsymbol{u}, \boldsymbol{v} \in \{-1, 1\}^n$, $\operatorname{err}$ is a semimetric.*

**Theorem 3.** *For $\phi_{in}, \phi_{out} \in [0,1]$, let $G$ be a $(2, \phi_{in}, \phi_{out})$-clusterable graph, and let $\eta < 1/2$. Then, any approximate algorithm with failure probability $\eta$ and misclassification rate $\zeta$ cannot satisfy $\epsilon$-DP for $\epsilon < 2\ln(1/(9e\zeta))/d$ on $d$-regular graphs.*

*Proof.* We follow a packing argument by Chen et al. (2023). Consider the set $S = \{\boldsymbol{p} \in \{-1,1\}^n \mid \mathbf{1}^\top \boldsymbol{p} = 0\}$ equipped with the semimetric err. Let $G$ be a balanced, $d$-regular, $(2, \phi_{in}, \phi_{out})$-clusterable graph with cluster membership vector $\boldsymbol{x_G} \in S$. Let $B = \mathcal{B}(x_G, 8\zeta)$. Let $P = \{\boldsymbol{x_G}, \boldsymbol{x_1}, \ldots, \boldsymbol{x_p}\}$ be a maximal $2\zeta$-packing of $\mathcal{B}$. For every $\boldsymbol{x_i} \in P$, there exists a $(2, \phi_{in}, \phi_{out})$-clusterable graph with cluster membership vector $\boldsymbol{x_i}$ and $\mathrm{err}_G(\boldsymbol{x_i}) \leq 6\zeta dn$: Since $\boldsymbol{x_i}$ is balanced, the number of vertices $j$ such that $(\boldsymbol{x_i})_j = -1$ and $(\boldsymbol{x_G})_j = 1$ is equal to the number of vertices $j'$ such that $(\boldsymbol{x_i})_{j'} = 1$ and $(\boldsymbol{x_G})_{j'} = -1$. Consider a perfect matching between these two sets of vertices and, for each matched pair, swap their neighbors in $G$ to obtain $H_{\boldsymbol{x_i}}$.

By the maximality of $P$, we have that $\mathcal{B}(\boldsymbol{x_G}, 6\zeta) \setminus \cup_i(\mathcal{B}(\boldsymbol{x_i}, 4\zeta)) = \emptyset$. Otherwise, we could extend $P$ by $\boldsymbol{x_{p+1}}$, where $\boldsymbol{x_{p+1}}$ is an element of the non-empty difference. This would contradict the maximality of $P$. Therefore, it follows that

$$\sum_{i=1}^{p} |B(\boldsymbol{x_i}, 4\zeta)| \geq |B(\boldsymbol{x_G}, 6\zeta)| \Rightarrow p \geq \frac{|B(\boldsymbol{x_G}, 6\zeta)|}{|B(\boldsymbol{x_G}, 4\zeta)|} \geq \frac{\binom{n/2}{3\zeta n}^2}{\binom{n/2}{2\zeta n}^2} \geq \frac{(\frac{1}{6\zeta})^{6\zeta n}}{(\frac{e}{4\zeta})^{4\zeta n}} \geq \left(\frac{2}{18e\zeta}\right)^{2\zeta n}.$$

By group privacy and an averaging argument, there exists $i \in [p]$ so that

$$\Pr[\mathcal{A}(H_{\boldsymbol{x_i}}) \subseteq \mathcal{B}(\boldsymbol{x_i}, \zeta)] \leq \exp(\epsilon\zeta nd)\Pr[\mathcal{A}(G_{\boldsymbol{x_G}}) \subseteq \mathcal{B}(\boldsymbol{x_i}, \zeta)] \leq \exp(\epsilon\zeta nd)\frac{\eta}{p}.$$

Now, assume that there is an $\zeta$-approximate, $\epsilon$-DP algorithm with failure probability $\eta$. By assumption, $1 - \eta \leq \Pr[\mathcal{A}(H_{\boldsymbol{x_i}}) \subseteq \mathcal{B}(\boldsymbol{x_i}, \zeta)]$. Rearranging, we obtain

$$\frac{2\ln\left(\frac{1}{9e\zeta}\right)}{d} \leq \frac{2\zeta n \ln\left(\frac{2}{18e\zeta}\right)}{\zeta dn} \leq \frac{\ln(1-\eta) + \ln\left(\left(\frac{2}{18e\zeta}\right)^{2\zeta n}\right) - \ln(\eta)}{\zeta dn} \leq \epsilon. \qquad \square$$

## E  DETAILS ON EXPERIMENTS

For reference and reproducibility, we provide some observations on the running times and scalability. The experiments were run on a single Google Cloud n2-standard-128 instance. The CPU time (single core running time) for one run was about 2 minutes. The number of runs we performed is 100 per input graph, times 10 input graphs, times 6 SBM parameterizations, times 2 algorithms, i.e., 12,000 runs. The running times roughly scale as follow as a function of the input size: for $n = 400$ about 12 minutes, for $n = 600$ about 195 minutes, for $n = 800$ about 790 minutes.

