# OpenReview forum: "A Differentially Private Clustering Algorithm for Well-Clustered Graphs"
_ICLR.cc/2024/Conference — ICLR 2024 poster_

### Official Review · Reviewer_viwC · 2023-10-31

**Soundness:** 4 excellent
**Presentation:** 3 good
**Contribution:** 4 excellent
**Rating:** 8
**Confidence:** 3

**Summary:**

The paper studies differentially private clustering algorithms for balanced and well-clustered graphs. The notion of ‘’well-clustered’’ is defined by a high ratio between the inner and the outer conductances, and the notion of ‘’balanced’’ is defined by the graph volume. The paper considers the problem with any fixed $k$ number of clusters, which is a natural extension of a recent result by CCDEIST [Neurips’23] that studied the DP clustering algorithms for balanced and well-clustered instances with 2 clusters. Instead of using a cost function, the goal in this setting is to ‘’almost entirely’’ recover the clustering labels while still preserving the differential privacy. This is a reasonably strong goal given the strong assumption on the underlying instances.

The main result of the paper is an $(\epsilon, \delta)$-DP algorithm that recovers the clusters correctly up to an $O(\log(1/\delta)/\epsilon^2)$ multiplicative factor on the *volume* of the (ground truth) cluster (assuming constant balance factor). The paper also provides a lower bound showing that we cannot hope for zero-error algorithms under this setting (unlike the exact recovery of CCDEIST [Neurips’23]). Finally, the paper conducts several experiments on synthetic datasets, and the results show that the proposed algorithm is competitive in the *utility* metric (but did not test for privacy).

The main technical ingredient of the paper is a white-box adaptation of the results of CCDEIST [Neurips’23]. In particular, the algorithm uses a variate of the $\ell_1$ sensitivity bound by CCDEIST [Neurips’23] for strongly convex functions and shows that their SDP objective function satisfies the desired convexity property. As such, they can get a levy on the level of Gaussian noise put on the output of the SDP solution. They further use known results in spectral clustering to show that the added noise level does not induce too much error on the utility.

**Strengths:**

My general evaluation of the paper is positive: it studies an interesting problem, which in turn is a natural follow-up of a well-established result. The paper is well-written, and the technical steps are rigorous and easy to follow. The organization of the paper also looks quite nice.

I checked the privacy proof, and things look correct to me, barring the details of the proof of Lemma 3.1. Unfortunately, due to the limits on the review timeline, I did not get a chance to verify the algebra for the utility proof – a spot-check indicates that nothing is wrong.

Finally, I think the results have nice applications due to the widespread popularity of the SBM model and the fact that a lot of networks satisfy the ``clustered’’ property. The experiment results show that the running time of the algorithm is not too bad, at least on small-scale of data.

**Weaknesses:**

Presentation problems in the main theorem. a). DP is not formally defined before the statement of theorem 1. Please at least add a pointer to your definition in section 2. b). The notation for $G_1 \Delta G_2$ is never defined. (And it’s never defined anywhere in the paper! :)) ) and c). The trade-off between the utility and the $(\epsilon,\delta)$-DP is not clear in the statement (although there is a lower bound on $m$), and it was unclear until Lemma 4.3. I would prefer to state the bound for general instances and factor the trade-off in the multiplicative error.

The notion of $(\kappa, D_{1}, D_{2})$-strongly convex is not defined in the preliminary (and only becomes clear to me after looking into the proof).

While I understand this is a mainly-theory paper, I would still suggest the authors spend more passages to properly motivate the study of the clustering of well-clustered graphs. I think CCDEIST [Neurips’23] has a nice example on this front by discussing the motivations from stochastic block models and downstream applications. Some discussion on this front is especially important for the broader appeal of the ICLR audiences.

Given the fact that the algorithm needs to run SDP, it’s unlikely that the algorithm can be scaled to modern social networks, which are the best candidates for well-clustered properties. Purely combinatorial algorithms would be much more helpful on this front.

**Questions:**

Defining balance with volume seems to be a strong notion since it rules out the case when the clusters have comparable sizes but different densities. In contrast, I believe setting of CCDEIST [Neurips’23] only requires the *vertices* in two clusters to be balanced. Do you have any comments on this front?

Can you report the runtime of your experiments and the hardware for your experiments? I want to get a sense of how far this algorithm is from being implementable in the real world and in the industry. (Please note that a bad performance will not affect my score, so please don’t do it strategically :)))

Your lower bound is for pure DP, can you get any result for pure $\epsilon$-DP upper bound with your technique? In particular, since you have a bound for the $\ell_{1}$ sensitivity of $f$, can you use Laplace mechanism to obtain a pure DP result?

**Details Of Ethics Concerns:**

I do not have any ethical concerns.

---

> ### Author Response · Authors · 2023-11-16
>
> We thank you for your reviews and address your concerns as follows.
>
> Q1: a). DP is not formally defined before the statement of theorem 1. b). The notation for $G_1 \Delta G_2$ is never defined. c). The trade-off between the utility and the $(\epsilon,\delta)$-DP is not clear in the statement.
>
> A1: We will incorporate your suggestions in the revised version. For example, we will add a pointer to the definition of DP before Theorem 1. The notion of $C_1\triangle C_2=(C_1\cup C_2)\setminus (C_1\cap C_2)$ is the symmetric difference between the sets $C_1$ and $C_2$.  We will add a remark on the bound for the general instances and add a corresponding pointer to Lemma 4.3.
>
> Q2: The notion of $(\kappa, D_{1}, D_{2})$-strongly convex is not defined in the preliminary.
>
> A2: We have given the definition in Definition C.2, which is included in the appendix due to space constraints. We will add a pointer to the definition in the main text.
>
> Q3: I would still suggest the authors spend more passages to properly motivate the study of the clustering of well-clustered graphs.
>
> A3: Thanks for the suggestion. We will add some motivations or examples as follows. Consider any social network (Facebook, LinkedIn) that is defined by all the users in Europe. Intuitively, all the users from the same country will form a large cluster, and the whole graph is a good candidate for a well-clustered graph (and the linkage information is naturally private).  One type of graph that often appears in downstream applications like recommendation or abuse detection applications is bipartite graphs (often these are social graphs). For example, such a graph may be defined on customers and products, and edges indicate that a customer purchased a product. Good clusters in these graphs contain nodes from both sides, but vanilla SBM models cannot generate these bipartite graphs with high probability. Studying a clustering property of graphs instead of a generative model allows us to overcome these shortcomings.
>
> Q4: Given the fact that the algorithm needs to run SDP, it’s unlikely that the algorithm can be scaled to modern social networks, which are the best candidates for well-clustered properties. Purely combinatorial algorithms would be much more helpful on this front.
>
> A4: We agree that it will be useful to have purely combinatorial algorithms, but currently we do not know how to obtain such algorithms with competitive privacy guarantee.
>
> Q5: Defining balance with volume seems to be a strong notion since it rules out the case when the clusters have comparable sizes but different densities. In contrast, I believe setting of CCDEIST [Neurips’23] only requires the vertices in two clusters to be balanced.
>
> A5: For graphs with varying degrees, it is much more convenient or natural to use volume to measure the importance of a vertex set $C$, which is roughly the number of all involved edges incident to $C$. This has already been discussed in the influential paper by Kannan, Vempala, and Vetta ("On clusterings: Good, bad and spectral." Journal of the ACM 2004), who motivated to use ''conductance'' to measure the quality of a cluster, instead of using the ''expansion''. The former is defined to be the ratio between the number of crossing edges and the *volume*, while the latter is the ratio between the number of crossing edges and the *size*. This leads to the use of volume to measure the importance of vertex sets in studying graph clustering.
> On the other hand, note that for graphs in which all the vertices have roughly the same degrees, it is almost the same to use the volume and sizes to measure the importance of a set or measure the balance. Under the assumptions of the SBM in CCDEIST[Neurips'23], the degrees of all vertices are roughly the same with high probability and their degrees cannot vary too much. Our algorithm deals with non-regular graphs, so adopting volume balance would be more appropriate.
>
> Q6: Can you report the runtime of your experiments and the hardware for your experiments? I want to get a sense of how far this algorithm is from being implementable in the real world and in the industry.
>
> A6: The experiments were run on a single Google Cloud n2-standard-128 instance. The CPU time (single core running time) for one run was about 2 minutes. For the experiments, we have 100 runs per input graph, times 10 input graphs, times 6 SBM parameterizations, times 2 algorithms, i.e., 12,000 runs.
>
> Q7: Your lower bound is for pure DP, can you get any result for pure $\epsilon$-DP upper bound with your technique? In particular, since you have a bound for the $\ell_1$ sensitivity of $f$, can you use Laplace mechanism to obtain a pure DP result?
>
> A7: Our current approach cannot get any result for pure $\epsilon$-DP, as we can only guarantee a good SDP solution with small $\ell_2$ sensitivity, due to the use of strong convexity of the regularized SDP. For $\epsilon$-DP, one always needs to have a non-trivial $\ell_1$-sensitivity bound.

---

> > ### Comment · Reviewer_viwC · 2023-11-20
> > **Responses + additional question**
> >
> > Thanks for the responses. I'm satisfied with the motivating example and the justification for using conductance according to the KVV paper. I would recommend adding the justification to your paper for later versions.
> >
> > For the experiment: since the running time is quite fast (2 mins) for the 100-vertex instance, I think it makes sense to test empirically the scalability of your algorithm, i.e., try something like $10^4$ and see how long it would take for the algorithm to terminate (or it just breaks down). Graphs of ~100 vertices are still too small for most applications, including the social networks you mentioned in the response.
> >
> > A follow-up question for $\epsilon$-DP: what is the barrier to proving a lemma that works for $\ell_1$ sensitivity with the SDP? In particular, can you elaborate more about A7? (I.e., technically, why can't one bound $\ell_1$ sensitivity for the function $g(G)$ in lemma 4.1?)

---

> > > ### Author Response · Authors · 2023-11-22
> > >
> > > Thank you very much for your further comments. We will add the justification as you suggested.
> > >
> > > Regarding the experiments, we further investigated the running time of larger instances, but we may not be able to evaluate the quality of significantly larger instances with the same statistical certainty as for n=100 / n=150 (as we did in the submission). The instances we tried had the following  running times: n=400 -> 12min, n=600 -> 195min, n=800 -> 790min.
> > >
> > > Regarding the question on $\epsilon$-DP: In Lemma 4.1, we added a regularization term, which is a Frobenius norm of (a scaled version of ) matrix $X$ (equivalently, the $\ell_2$-norm of the corresponding $n^2$-dimensional vector), in the definition of the objective function $f(G,X)$. This ensures that the function $f(G,X)$ is strongly convex. As observed in [CCDEIST], if the two strongly convex functions over constrained sets are point-wise close, then their minimizers are also close, which essentially tells us that we can have a good bound on the $\ell_2$-sensitivity of $g(G)$.
> > >
> > > To obtain the $\ell_1$-sensitivity of $g(G)$, one can try to directly apply the relation between $\ell_1$-norm and $\ell_2$-norm of a vector, but this only gives a bound of $\sqrt{24(\lambda+3)m}\cdot n$, which implies that to get $\epsilon$-DP, we need to add too much noise and the resulting utility bound will be meaningless.
> > >
> > > On the other hand, one may be tempted to add the $\ell_1$-norm of (some scaled version of) $X$ as a regularization term, but then we cannot guarantee the strong convexity property of the resulting function $f$, and even if $f$ is still point-wise close, we do not know how to translate it into a $\ell_1$-sensitivity bound for the minimizers.

---

### Official Review · Reviewer_vyy9 · 2023-10-31

**Soundness:** 4 excellent
**Presentation:** 4 excellent
**Contribution:** 3 good
**Rating:** 6
**Confidence:** 3

**Summary:**

This paper studies differentially private algorithms for cluster recovery on well-clusterable graphs. A well clusterable graph is an unweighted graph that has a bounded ratio between outer conductance and inner conductance. The notion of privacy is (eps, delta)-differntial privacy, and it is measured on neighboring graphs that have one edge difference. The goal is to minimize the (relative) number of points that are mis-clustered in every cluster, provided that the input graph is well clusterable into k clusters (and assuming k is known).

The main result achieves a misclassification ratio only slightly worse than the state-of-the-art non-private version with respect to the main parameters. The price of privacy is mostly on the requirement of the desntiy of graphs, i.e., the number of edges must be at least some poly(\phi_in / \phi_out) / eps^2 * log(1 / \delta) factor larger than the number of vertices. Lower bounds were also obtained to justify this price of privacy. Finally, some experiments were provided to justify the usefulness of the new algorithm.

**Strengths:**

- The bound looks strong since it is very close to the non-private version — The privacy guarantee does not directly add to the misclassification ratio. The factor of poly(phi_in / phi_out) in the “price of privacy” seems to be natural and is probably necessary (even though the paper did not justify this).

- The techniques are not particularly novel, but combining existing techniques and obtaining interesting results is also a solid contribution.

- The paper is well-written and easy to read.

**Weaknesses:**

- One thing I’m not sure: does the balance parameter c also appear in the bound of Peng et al. 2015? This is not clearly discussed in your paper, and I think that this is very important.

- Whether (eps, delta)-DP is necessary is not justified; the upper bound is (eps, delta)-DP and the lower bound is about eps-DP.

- The approach requires to solve SDP which is heavy for larger datasets.

- The experiment is not comprehensive and I find it not very convincing.

**Questions:**

- The presentation is technically heavy. It would be helpful to talk about more intuitions and less calculations in the main text.

- In Theorem 1, n and m are not quantified. Also, the quantification of \sigma is not very clear — “for every” or “exists”?

- In the experiments, it would be nice to define or at least explain what AMI and NMI are, and what do they indicate. I’m not even sure if larger value is better.

- I found only arXiv version is cited for several papers — are they published in a conference? Please check and update.

- You mentioned that your experiments are to somehow verify that “when the noise added to the adjacency matrix by randomized response is balancing with its signal, the output of Algorithm 1 still has significant utility”, but I don’t see this clearly discussed. For instance, from what data did you see that the noise added by randomized response is balancing with its signal? And how exactly does your Algorithm 1 compare w.r.t. utility?

- I may have missed something, but did you say the value of eps and delta that you use in the experiments?

---

> ### Author Response · Authors · 2023-11-16
>
> We thank you for your reviews and address your questions as follows.
>
> Q1: One thing I’m not sure: does the balance parameter c also appear in the bound of Peng et al. 2015? This is not clearly discussed in your paper, and I think that this is very important.
>
> A1: Peng et al. did not explicitly specify the need for balance. Thank you for pointing this out, and we will incorporate these discussions.
>
> Q2: Whether (eps, delta)-DP is necessary is not justified; the upper bound is (eps, delta)-DP and the lower bound is about eps-DP.
>
> A2: Our Theorem 2 essentially indicates that when the classification error is $o(1)$, there is no pure-DP algorithm for this problem with a constant $\epsilon$. We will provide further clarification on this in the paper. On the other hand, our algorithmic result Theorem 1 says for constant epsilon and sufficiently small delta, one can achieve classification error $o(1)$, which is impossible for pure-DP algorithm. We will add this explanation in the revised version.
>
> Q3: The approach requires to solve SDP which is heavy for larger datasets.
> Q4: The experiment is not comprehensive and I find it not very convincing.
>
> A3, A4: When the datasets are large, repeatedly solving the SDP to obtain statistically significant findings does take a long time, which is also why we did not conduct experiments on larger datasets.
>
> Q5: The presentation is technically heavy. It would be helpful to talk about more intuitions and less calculations in the main text.
>
> A5: We will add more intuitive discussion on the formalization of SDP and some lemmas, and add intuitive comparison between the $(\epsilon,\delta)$-DP algorithm and the $(\epsilon,0)$-DP lower bound, etc.
>
> Q6: In Theorem 1, n and m are not quantified. Also, the quantification of $\sigma$ is not very clear — ''for every'' or ''exists''?
>
> A6: We will clearly state that Theorem 1 holds for "n-vertex and m-edge graph G" and explicitly point out that "there exists $\sigma$".
>
> Q7: In the experiments, it would be nice to define or at least explain what AMI and NMI are, and what do they indicate. I’m not even sure if larger value is better.
>
> A7: In probability theory and information theory, the mutual information (MI) of two random variables is a measure of the mutual dependence between the two variables. AMI and NMI are two different forms of MI. Indeed, the larger the values are, the better the performances are. We will add this in the revised version.
>
> Q8: I found only arXiv version is cited for several papers — are they published in a conference? Please check and update.
>
> A8: Thanks for pointing this out. Indeed, some references have been published at conferences. For example, the cited Chen et al's work is published at NeurIPS 2023, and the cited Liu et al's work is published at COLT 2022. We will update these citations.
>
> Q9: You mentioned that your experiments are to somehow verify that ''when the noise added to the adjacency matrix by randomized response is balancing with its signal, the output of Algorithm 1 still has significant utility'', but I don’t see this clearly discussed. For instance, from what data did you see that the noise added by randomized response is balancing with its signal? And how exactly does your Algorithm 1 compare w.r.t. utility?
>
> A9: For $p$ between $0.2$ and $0.3$ and $q$ between $0.0$ and $0.1$, randomized response will, in expectation, blur the cluster and non-cluster edge densities. For example, for $p=0.2$ and $q=0$, the expected (empirical) $p'$ and $q'$ after randomized response (with $p_{RR}=0.27$) would be $0.36$ and $0.27$, so $|p'-q'|=0.09$. Considering $|p-q|$ as the strength of the clustering signal in the original graph, roughly half of it is shadowed by randomized response in the noisy graph. We agree that ''balancing'' for $|p-q|/2 = |p'-q'|$ is a sloppy term. We initially intended it as an exposition and motivation for the experiment, but we will add a more technical explanation. The utility is measured by AMI and NMI with respect to the ground truth clusters of the SBM. We will clarify this in the revised version as well.
>
> Q10: I may have missed something, but did you say the value of eps and delta that you use in the experiments?
>
> A10: Yes, please note that we mentioned it in ''Datasets and setup.'': $\epsilon = 1$ and $\delta = 1/n^2$.

---

### Official Review · Reviewer_ApBn · 2023-11-01

**Soundness:** 3 good
**Presentation:** 2 fair
**Contribution:** 3 good
**Rating:** 6
**Confidence:** 2

**Summary:**

The paper gives a DP algorithm for well-clustered graphs. Here a "well-clustered graph" is similar to graphs generated from some stochastic block model, such that there exists an underlying partition of the graph, where edges between clusters are sparse (i.e, small outer-conductance) while each cluster internally is well-connected (large inner-conductance). The task is to recover this underlying partition in a differentially-private way.
The paper gives a spectral algorithm for the problem: first solve a SDP relaxation of the problem to extract the cluster structure, where (the assignment of) each vertex is embedded as a k-dimensional vector, then apply standard Gaussian DP scheme to (the Gram matrix of) the embedding vectors, which gives a perturbed version of the embedding. The final clustering is obtained by standard k-means clustering over the perturbed vectors.
The paper gives theoretical analysis on the algorithm performance, measured by the volume of the misclassified vertices, and prove it satisfy DP requirement.

**Strengths:**

- The paper gives the first DP algorithm for a hard (?) problem. The algorithm achieves good error bound and utility that are close to best-known non-private algorithms.
- The algorithm given is very simple (maybe except for the SDP, whose formulation I believe lacks sufficient explanation), while the analysis seems quite non-trivial.

**Weaknesses:**

- I'm not familiar with spectral methods, so I'm not going to comment on technical merits. Although I can follow most proofs, I find it difficult to grasp the intuitions. One example is the SDP with the constraint (1). It would be much more reader-friendly if the authors can give intuitive summary of the ideas instead of directly jumping to proofs.

**Questions:**

- I'm curious what's the relation of this well-clustered graph with stochastic block model? The SBM seems to attract much more research interests. Is the problem studied here a generalization of SBM, or a special case, or neither?
- It would be very helpful if the authors can give some intuition & motivation of the SDP, in particular how the regularizer and the constraint (1) are designed.
- I'm curious why the DP noise is applied to the Gram matrix X rather than directly to vectors associated with each vertex?
- Proof of Lemma 4.4: I think the upper bound on $\sum_{i=1}^k|E(C_i,V-C_i)|$ is $\phi_\text{out}\cdot2m$, not $\phi_\text{out}\cdot m/2$.
- Proof of Lemma 4.4: Could you give a detailed explanation on how the second smallest eigenvalue $\mu$ of $L_{G\{C_i\}}can be obtained by minimizing the quotient given in the paper.
- In experiment the baseline is random response + SDP  *without regularizer*. Why do you remove the regularizer?

---

> ### Author Response · Authors · 2023-11-16
>
> We thank you for your reviews and address your questions as follows.
>
> Q1: I'm curious what's the relation of this well-clustered graph with stochastic block model? The SBM seems to attract much more research interests. Is the problem studied here a generalization of SBM, or a special case, or neither?
>
> A1: In general, SBM is a generative model, and well-clusterability is a structural property of graphs. In particular, edges in a well-clustered graph are not necessarily chosen independently with certain probabilities but can be added in an ''adversarial'' way. SBM can be parameterized so that it produces well-clustered graphs with high probability. On the other hand, there exist well-clusterable graphs that cannot be generated with high probability by any SBM parameterization (e.g., bipartite clusters, or clusterable graphs with constant average degree). In general, clustering on well-clustered graphs poses a more challenging problem.
>
> Q2: It would be very helpful if the authors can give some intuition & motivation of the SDP, in particular how the regularizer and the constraint (1) are designed.
>
> A2: The intuition of the SDP is the following. We would like to ensure a significant sum of vector distances for all pairs of points, and the objective is to minimize the vector distances between endpoints of all edges, thereby achieving a configuration where inter-class vector distances are large and intra-class vector distances are small. Considering a ground truth clustering $C_1, \cdots, C_k$ in a well-clusterable graph and a set of orthogonal bases $v_1, \cdots, v_k$. We can set the vectors associated with vertex $u$ to be $v_i$ if $u \in C_i$, then the objective can be upper bound by $2\phi_{out}m+\frac{8m}{\lambda}$, which reflects the number of edges crossing different clusters and the added regularizer part. The regularizer is added to ensure the strong convexity of the objective function, which in turn guarantees that the solution to the SDP is stable. We will add this discussion in the revised version.
>
> Q3: I'm curious why the DP noise is applied to the Gram matrix X rather than directly to vectors associated with each vertex?
>
> A3: The main reason is that we can only guarantee that the $\ell_2$-sensitivity of the matrix $X$ is small, or the dot products of vectors have small sensitivity. This is due to the fact that the objective function in SDP is strongly convex in $X$, which in turn guarantees that $X$ has small sensitivity by Lemma 3.1. On the other hand, we do not know how to bound the sensitivity of vectors.
>
> Q4: Proof of Lemma 4.4: The upper bound on $\sum_{i=1}^k|E(C_i,V-C_i)|$ is $\phi_\text{out}\cdot 2m$, not $\phi_\text{out}\cdot m/2$.
>
> A4: Thank you for pointing out this, we will fix this typo.
>
> Q5: Proof of Lemma 4.4: Could you give a detailed explanation on how the second smallest eigenvalue $\mu$ of $L_{G_{C_i}}$ can be obtained by minimizing the quotient given in the paper.
>
> A5: This is actually a known fact. That is, if $\mu$ is  the second smallest eigenvector of the *normalized* Laplacian matrix, by its relation to the Rayleigh quotient, it can equivalently be written as the formula that we used in Lemma 4.4. This fact can be found in the book "Spectral Graph Theory" by Fan Chung (page 5, equation 1.5). We will add this reference in the revised version. (We will add the missing word *''normalized''* in front of ''Laplacian matrix''. )
>
> Q6: In experiment the baseline is random response + SDP without regularizer. Why do you remove the regularizer?
>
> A6: The regularizer facilitates a trivial solution on the SBM to smooth and bound privacy. Since the input graph of the SDP is already DP after applying randomized response, the regularizer is not needed and would only worsen the quality of the result.

---

### Official Review · Reviewer_ZCFz · 2023-11-01

**Soundness:** 3 good
**Presentation:** 3 good
**Contribution:** 2 fair
**Rating:** 6
**Confidence:** 3

**Summary:**

This submission studies differentially private (DP) algorithms for recovering clusters in well-clustered graphs, and it proposes a private algorithm that works for well-clustered graphs with k nearly-balanced clusters. The authors also show that any (pure) ϵ-DP algorithm would result in substantial error.

**Strengths:**

1. The results seem interesting: The private algorithm in this submission works for
k nearly balanced clusters while the previous reference only works for 2.

2. The empirical results look promising, in comparison between RR+SDP and Algorithm 1.

**Weaknesses:**

1. The condition "WELL-CLUSTERED GRAPHS" may reveal sensitive information in edge privacy.

**Questions:**

1. The authors claim that this submission is inspired by a recent work of Chen et al, so what is the main difference between this submission and Chen et al's paper in the technique aspect?

2. For two adjacent data sets (graph) D and D' differing in one edge, could it happen that D is well-clustered while D' is not?

---

> ### Author Response · Authors · 2023-11-16
>
> We thank you for your reviews and address your questions as follows.
>
> Q1: The condition "WELL-CLUSTERED GRAPHS" may reveal sensitive information in edge privacy.
>
> A1: The privacy guarantee in Theorem 1 holds for any graph i.e., also not well-clustered graphs. The reason is that the algorithmic framework we are using ensures that upon small changes of the input graph, the output of the algorithm does not change significantly, *regardless of the input's nature*. Therefore, the algorithm does not leak private information with respect to $(\epsilon,\delta)$-DP. Only the utility guarantee depends on the well-clusterable property.
>
> Q2: The authors claim that this submission is inspired by a recent work of Chen et al, so what is the main difference between this submission and Chen et al's paper in the technique aspect?
>
> A2: In the following, we abbreviate Chen et al.'s work as [CCDEIST]. Since [CCDEIST] focuses on SBM with $2$ clusters and we are focusing on well-clustered graphs with $k$ clusters, the SDPs in these two works are different. Furthermore, [CCDEIST] uses the signs of the first eigenvector of the noisy version of the SDP solution for the rounding and uses the probabilistic tools for the analysis, while our rounding is based on the spectral embedding and the k-means algorithm, and our proofs predominantly employ spectral graph theory.
>
> Q3: For two adjacent data sets (graph) D and D' differing in one edge, could it happen that D is well-clustered while D' is not?
>
> A3: Yes. Since for every well-clustered graph $G$ and not well-clustered graph $H$ (of the same size) there is a sequence of edge insertions and removals that transforms $G$ into $H$, there exist such two adjacent pairs $D, D'$ in the sequence. Indeed, it can happen that in a well-clustered graph, one cluster (expander) might contain a dangling edge, whose removal will cause the graph to be not well-clustered. However, note that the privacy guarantee in Theorem 1 holds for any graph, regardless if the input graph is well-clustered or not.

---

### Meta-Review · Area_Chair_8p6r · 2023-12-05

**Metareview:**

The paper studies differentially private algorithms for recovering clusters in well-clustered graphs. Previous work solved the problem for the stochastic block model and only 2 clusters. The current work generalizes to k clusters and from the stochastic block model to a more general model. The algorithm is based on SDP and achieves the strong approximation guarantee that is qualitatively close to the non-private case. All reviewers are appreciative of the theoretical contribution and recommend acceptance.

On the other hand, some reviewers note that the use of SDP prevents the application to large graphs. Furthermore, the work only applies to balanced (similarly sized) clusters whereas the non-private algorithms seem to not require this condition.

**Justification For Why Not Higher Score:**

The use of SDP might limit the application to only moderate sized instances. Furthermore, the algorithm is only applicable to balanced instances.

**Justification For Why Not Lower Score:**

All reviewers appreciate the contribution to a well studied problem in the non-private setting, now with privacy and it significantly generalizes previous works.

---

### Decision · Program_Chairs · 2024-01-16

Accept (poster)